# Kilohertz transcranial magnetic perturbation (kTMP) as a new non-invasive method to modulate cortical excitability

**Ludovica Labruna[1,2,3]\*†, Christina Merrick[1,2]†, Angel V Peterchev[4,5,6,7], Ben Inglis[3], Richard B Ivry[1,2,3], Daniel Sheltraw[1,3]**

[1]Magnetic Tides, Inc, El Cerrito, United States; [2]Department of Psychology, University of California, Berkeley, Berkeley, United States; [3]Helen Wills Neuroscience Institute, University of California, Berkeley, Berkeley, United States; [4]Department of Psychiatry and Behavioral Sciences, Duke University, Durham, United States; [5]Department of Biomedical Engineering, Duke University, Durham, United States; [6]Department of Electrical and Computer Engineering, Duke University, Durham, United States; [7]Department of Neurosurgery, Duke University, Durham, United States

**\*For correspondence:**
lulabrun@gmail.com

†These authors contributed equally to this work

## eLife assessment

This **important** study introduces and evaluates the efficacy of a novel form of non-invasive brain stimulation in humans: kilohertz transcranial magnetic perturbation (kTMP). The evidence provided for the ability of kTMP to increase cortical excitability with minimal sensation is **compelling**, with two separate replication experiments. Although exploratory in nature, this work represents new avenues for non-invasive brain stimulation research that has potential long-term appeal for both clinical and research applications. This paper will be of significant interest to neuroscientists interested in brain stimulation.

**Abstract** Non-invasive brain stimulation (NIBS) provides a method for safely perturbing brain activity, and has been employed in basic research to test hypotheses concerning brain–behavior relationships with increasing translational applications. We introduce and evaluate a novel subthreshold NIBS method: kilohertz transcranial magnetic perturbation (kTMP). kTMP is a magnetic induction method that delivers continuous kHz-frequency cortical electric fields (E-fields) which may be amplitude modulated to potentially mimic electrical activity at endogenous frequencies. We used transcranial magnetic stimulation to compare the amplitude of motor-evoked potentials (MEPs) in a hand muscle before and after kTMP. In Experiment 1, we applied kTMP for 10 min over motor cortex to induce an E-field amplitude of approximately 2.0 V/m, comparing the effects of waveforms at frequencies of 2.0, 3.5, or 5.0 kHz. In Experiments 2 and 3, we used two forms of amplitude-modulated kTMP (AM kTMP) with a carrier frequency at 3.5 kHz and modulation frequencies of either 20 or 140 Hz. The only percept associated with kTMP was an auditory tone, making kTMP amenable to double-blind experimentation. Relative to sham stimulation, non-modulated kTMP at 2.0 and 3.5 kHz resulted in an increase in cortical excitability, with Experiments 2 and 3 providing a replication of this effect for the 3.5 kHz condition. Although AM kTMP increased MEP amplitude compared to sham, no enhancement was found compared to non-modulated kTMP. kTMP opens a new experimental NIBS space inducing relatively large amplitude subthreshold E-fields able to increase cortical excitability with minimal sensation.

## Introduction

Electromagnetic non-invasive brain stimulation (NIBS) refers to a group of methods that perturb brain function by coupling an applied electric field (E-field) to neurons without the need to introduce electrodes into brain tissue. The NIBS E-field can safely manipulate neural excitability, providing neuroscientists with a powerful tool to advance our understanding of brain function. Evidence that NIBS can promote brain plasticity (*Polanía et al., 2018*) has prompted clinicians to pursue NIBS interventions in the treatment of psychiatric and neurologic disorders (*Di Lazzaro et al., 2021*; *Fitzgerald, 2020*; *Grigoras and Stagg, 2021*; *Iglesias, 2020*; *Lefaucheur et al., 2020*; *Maatoug et al., 2021*).

The NIBS E-field amplitude can be categorized as subthreshold or suprathreshold. Suprathreshold fields are sufficient to elicit immediate action potentials in neurons initially at resting membrane potential. Subthreshold E-fields are insufficient to directly cause action potentials but are employed to alter the state of the targeted neurons on time scales ranging from immediate entrainment effects to plasticity effects that extend well past the stimulation epoch (*Huang et al., 2017*; *Liu et al., 2018*). As such, subthreshold and suprathreshold methods have different experimental and clinical utilities.

Two broad categories of NIBS methods exist, delivering the E-field to the brain via injection of current through electrodes in contact with the scalp or magnetic induction from a coil placed over the scalp. In electrical methods, such as transcranial electrical stimulation (tES), the current and E-field can be constant as in transcranial direct current stimulation (tDCS) or time-varying as in transcranial alternating current stimulation (tACS). Magnetic induction methods, such as transcranial magnetic stimulation (TMS), deliver a time-varying current to the coil, generating a changing magnetic field and consequently an induced E-field in superficial brain regions. TMS is delivered as a brief pulse (typically 200–300 µs) and is referred to as repetitive TMS (rTMS) when delivered as a train of pulses.

The E-fields of tES and TMS differ in important ways. First, for TMS, the E-field amplitude is linearly proportional to the frequency of the current source, whereas the tES E-field amplitude is independent of this frequency. Second, the E-fields for the two methods exist in orthogonal subspaces (*Sheltraw et al., 2021*). As such, the E-fields between TMS and tES cannot be matched exactly; for example,

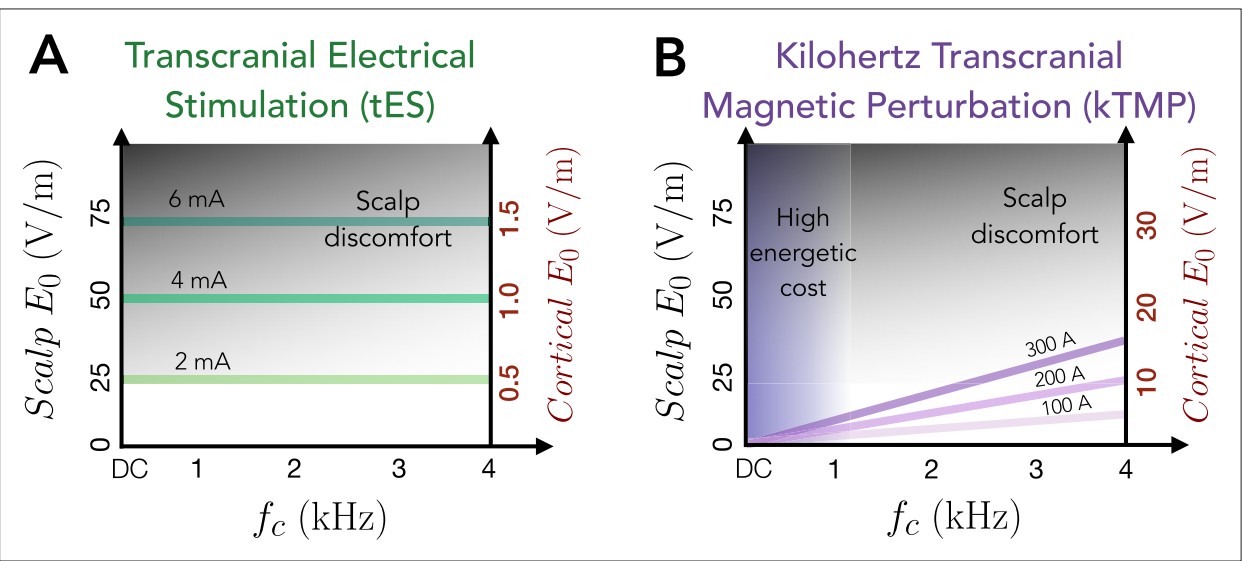

**Figure 1.** An illustration of the practical constraints of transcranial electrical stimulation (tES) and kilohertz transcranial magnetic perturbation (kTMP) in frequency and amplitude space. Solid lines represent the dependency of the electric field (E-field) amplitude upon frequency and amplitude of the electric current supplied to the electrodes (**A**) or coil (**B**). Calculations are based on a typical 5 × 7 cm tES electrode montage or figure-of-eight transcranial magnetic stimulation (TMS)/kTMP coil. The left vertical axis represents estimated scalp E-field amplitude while the right vertical axis represents the estimated motor cortex E-field amplitude. Shaded zones represent regions of the perturbation space constrained by practical considerations. Both methods are constrained by the scalp E-field magnitude, which at high values may result in discomfort due to peripheral nerve stimulation (gray shading). Note the substantial difference in the cortical E-field range that can be delivered tolerably for tES and kTMP. Illustrated here are approximate levels of discomfort for sustained waveforms (tES and kTMP) as opposed to pulsed methods (e.g., TMS). Note that magnetic induction methods such as kTMP (or TMS) are additionally constrained by high energetic costs (purple shading) required to generate E-fields of sufficient magnitude to influence neuronal states at low frequencies.

the TMS E-field does not have a radial component. Thus, even when the E-fields for tES and TMS are similar in focality, they may target different populations of neurons.

Third, for spatially similar cortical E-fields, the scalp E-field amplitude for tES is much greater than the scalp E-field for TMS because a large fraction of tES current travels along the path of least resistance and is, thus, shunted across the scalp (*Liu et al., 2018*). Although dependent on a number of factors, this property can be captured by the ratio of scalp to cortical E-field, which is approximately 18 times larger with tES compared to TMS (*Sheltraw et al., 2021*). Thus, the scalp E-field amplitude ultimately places a severe constraint on the focality and amplitude of tES cortical E-fields due to scalp peripheral nerve stimulation and muscle stimulation. Estimates of the tES cortical E-fields in the physiological frequency range suggest that the maximum for most human participants is around 0.5 V/m (*Huang et al., 2018*; *Vöröslakos et al., 2018*); beyond this value, scalp stimulation becomes detectable and soon intolerable. TMS systems are far less burdened by constraints imposed by the scalp E-field amplitude, allowing the method to be used to produce both subthreshold and suprathreshold cortical E-fields.

Motivated by these considerations, we developed a new magnetic induction NIBS approach, in which the subthreshold modulation of neural excitability is brought about via oscillating magnetic fields at kHz frequencies. We refer to this system as kilohertz transcranial magnetic perturbation (kTMP). In our new method, we drive a TMS coil using a high-current large-bandwidth amplifier operating in the kHz range. In this way, the E-field magnitude constraints imposed by scalp stimulation associated with tES are greatly diminished (*Figure 1*). The energetic requirements for a magnetic induction system would be prohibitive at low frequencies such as those measured with EEG. However, this cost is reduced to a manageable level by operating at higher frequencies. While the prototype kTMP system used in the present experiments produced cortical E-fields of 2 V/m at 2–5 kHz, kTMP has the potential to significantly increase the range of subthreshold E-field induction with the focality typical of TMS systems. In addition, the kTMP waveform can be amplitude modulated at frequencies matching endogenous neural rhythms (*Esmaeilpour et al., 2021*).

Since neurons near resting membrane potential act as low-pass filters, a critical question centers on whether narrow band kHz E-fields can effectively couple to the transmembrane potential to influence neural activity. This question has been explored in invasive and non-invasive empirical studies (reviewed in *Neudorfer et al., 2021*) and in model simulations (*Wang et al., 2023*). Experimentally, suprathreshold kHz fields have been used to produce nerve blocking (*Bhadra et al., 2018*) and temporal interference effects (*Grossman et al., 2017*). Converging evidence also indicates that subthreshold kHz fields are effective in modulating neural excitability. In particular, kHz tACS can induce an increase in post-stimulation motor-evoked potentials (MEPs) approximately equal to that observed following standard, low-frequency tES or rTMS (*Chaieb et al., 2011*). Intriguingly, low-frequency magnetic stimulation, a kHz method that mimics the magnetic field waveforms used in MRI, applies weak kHz E-fields across most of the brain and has been reported to have mood-altering effects (*Rohan et al., 2014*; *Carlezon et al., 2005*; *Rohan et al., 2004*; *Dubin et al., 2019*). In addition, kHz E-fields, again applied by MRI gradients, have been shown to alter brain glucose metabolism in a manner that scales with the field amplitude (*Volkow et al., 2010*).

We report here the results of three experiments with healthy human participants that evaluate the efficacy of kTMP in modulating cortical excitability. Adopting a conventional approach for evaluating the efficacy of NIBS methods (*Huang et al., 2005*; *Nitsche et al., 2005*; *Nitsche and Paulus, 2001*; *Nitsche and Paulus, 2000*), we used suprathreshold TMS to measure MEPs elicited in a hand muscle, comparing the amplitude of the MEPs before and after kTMP stimulation. For all the experiments, the kTMP amplitude was set to produce a peak cortical E-field amplitude of approximately 2.0 V/m at the targeted primary motor cortex. In Experiment 1, we tested non-amplitude modulated kTMP at three different carrier frequencies (2, 3.5, and 5 kHz), comparing these conditions to a sham condition (0.01 V/m at 3.5 kHz). In Experiments 2 and 3, the carrier frequency was fixed at 3.5 kHz and we investigated two forms of amplitude-modulated kTMP (AM kTMP) with modulation frequencies of either 20 or 140 Hz.

# Results

Across the three experiments, participants found kTMP stimulation to be tolerable and, indeed, as described below, essentially imperceptible. No adverse events occurred, and no discharges were observed in the electromyography (EMG) after kTMP stimulation.

In each experiment, five TMS probe blocks were sandwiched around active or sham kTMP stimulation (Figure 7A). We pooled the data across the two pre-kTMP probe blocks to establish a baseline measure. The post-stimulation blocks assessed changes in neural excitability at three time points after kTMP stimulation, operationalized as average percent change post kTMP relative to baseline.

## Single-pulse assay

### Repeated kTMP conditions (3.5 kHz and sham) across experiments

In Experiment 1, three carrier frequencies of kTMP stimulation were tested (2, 3.5, and 5 kHz) along with a sham kTMP condition (0.01 V/m, 3.5 kHz). Experiments 2 and 3 included the same non-modulated 3.5 kHz condition (and sham), providing an opportunity for evaluating reproducibility and a reference point for assessing the effect of AM kTMP (see below).

We first tested if the change in corticospinal excitability for the 3.5 kHz condition varied across the three experiments. Using a mixed-effects model with fixed factor of Experiment and random effect of participant, we found no difference across the three experiments for the non-modulated 3.5 kHz condition [$\chi^2(2)$ = 0.41, p = 0.813]. Similarly, we found no difference across experiments for the sham condition [$\chi^2(2)$ = 2.77, p = 0.251; *Figure 2A*]. *Figure 2B* displays change in MEP amplitude across the three post-kTMP blocks for sham and the 3.5 kHz condition for the three experiments. Based on these results, we combined the data across the three experiments for these two conditions in subsequent analyses. Note that, although the factor of experiment was not significant, we include experiment as a fixed factor in subsequent models that include data from the three experiments.

### Non-modulated kTMP increases cortical excitability

We next compared sham vs. active non-modulated kTMP (2, 3.5, and 5 kHz) and found that active kTMP produced a significant increase in corticospinal excitability [$\chi^2(1)$ = 23.83, p < 0.001; *d* = 0.51]. Pairwise comparisons of each active condition to sham showed that an increase was observed following both 2 kHz [$\chi^2(1)$ = 6.90, p = 0.009; *d* = 0.49] and 3.5 kHz kTMP [$\chi^2(1)$ = 37.57, p < 0.001; *d* = 0.70; *Figure 3*: Non-Modulated conditions]. The 5 kHz condition failed to reach significance [$\chi^2(1)$ = 1.43, p = 0.232; *d* = 0.21].

A comparison of the three active conditions showed that the effect on MEP amplitude was influenced by carrier frequency [$\chi^2(1)$ = 9.60, p = 0.008]. Specifically, the 3.5 kHz condition produced a larger increase in MEP amplitude compared to the 5 kHz condition [$\chi^2(1)$ = 8.64, p = 0.003; *d* = 0.50]. No difference was observed between the 2 and 3.5 kHz conditions [$\chi^2(1)$ = 3.25, p = 0.071; *d* = 0.30] or between the 2 and 5 kHz conditions [$\chi^2(1)$ = 2.44, p = 0.118; *d* = 0.28].

The effect of active non-modulated kTMP stimulation persisted across 36 min of the post-stimulation probe blocks. There was no significant effect of Time (i.e., Post Block) on MEP amplitude [$\chi^2(2)$ = 5.778, p = 0.056] nor was there a significant interaction between Time and Stimulation Condition for non-modulated kTMP [$\chi^2(6)$ = 109.0696, p = 0.1226; *Figure 3*: Non-Modulated conditions].

### AM kTMP produces similar increase in cortical excitability

We tested the effect of AM kTMP in Experiments 2 and 3, with the two experiments using different forms of amplitude modulation ($E_{AM1}$ and $E_{AM2}$; see Figure 8). For these experiments, we used a carrier frequency of 3.5 kHz given that this condition had produced the largest effect in Experiment 1.

Similar to what we observed from the single-pulse (SP) probe for non-modulated 3.5 kHz stimulation, AM kTMP resulted in an increase in MEP amplitude compared to sham [$\chi^2(1)$ = 23.59, p < 0.001; *d* = 0.54; *Figure 3*, AM conditions]. Pairwise comparisons showed that this effect was observed in three of the four AM conditions: $E_{AM1}$ [$f_m$ = 20, $f_b$ = 40; $\chi^2(1)$ = 13.72, p < 0.001; *d* = 0.74], $E_{AM2}$ [$f_m$ = 140, $f_b$ = 140; $\chi^2(1)$ = 5.61, p = 0.018; *d* = 0.62], and $E_{AM1}$ [$f_m$ = 140, $f_b$ = 280; $\chi^2(1)$ = 22.69, p < 0.001; *d* = 1.00]. The $E_{AM2}$ condition [$f_m$ = 20, $f_b$ = 20] showed a numeric increase, but this comparison was not significant [$\chi^2(1)$ = 2.75, p = 0.097; *d* = 0.427].

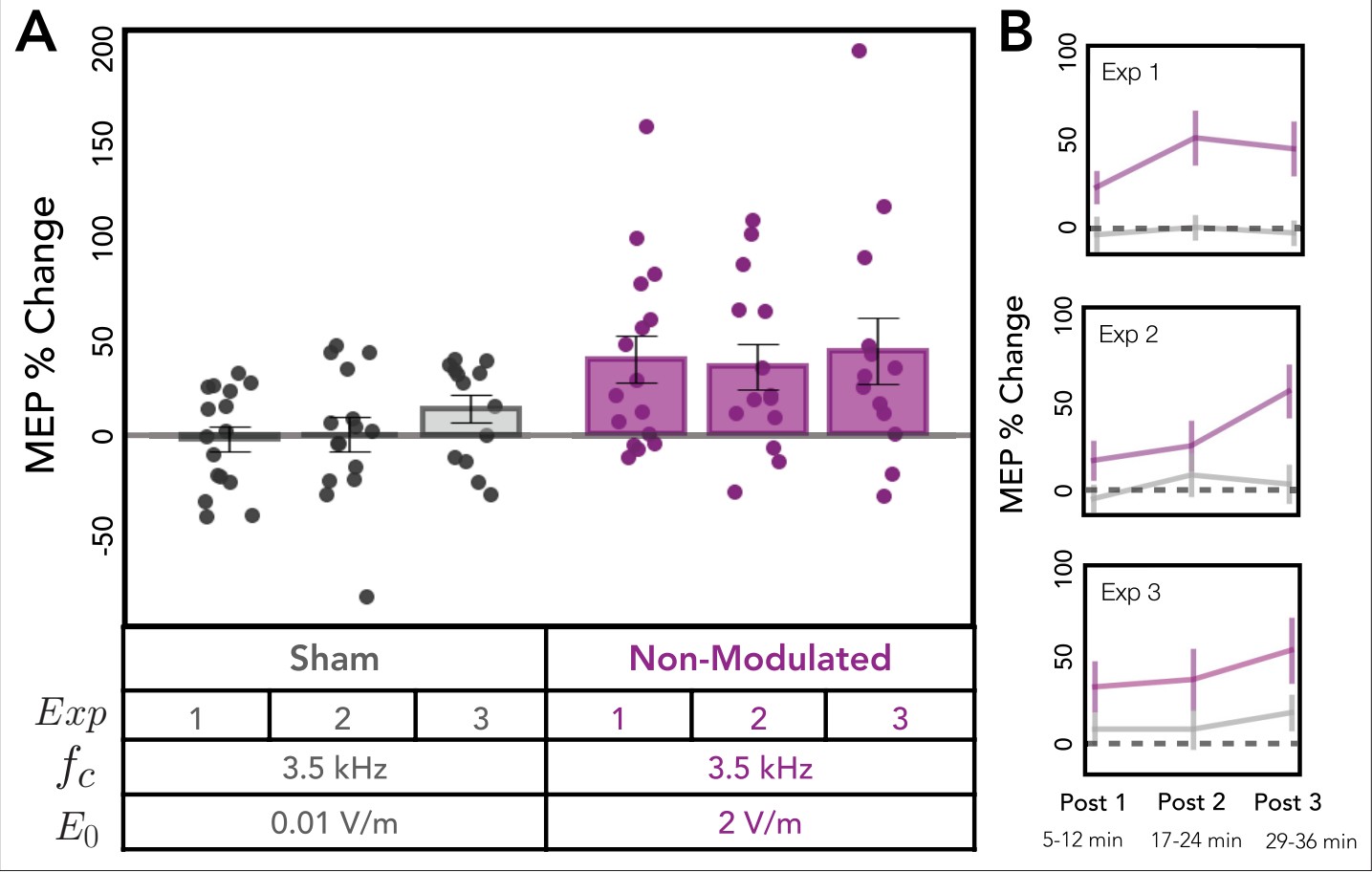

**Figure 2.** Replication of post-stimulation change in cortical excitability for sham and non-modulated 3.5 kHz kilohertz transcranial magnetic perturbation (kTMP) stimulation. (**A**) Change in motor-evoked potential (MEP) amplitude measured with single-pulse transcranial magnetic stimulation (TMS) following sham stimulation (left) and kTMP stimulation at 3.5 kHz (right). Dots denote values for individual subjects, bars—mean values, and whiskers—standard error. MEP change post-intervention did not significantly differ across experiments for sham and 3.5 kHz. (**B**) Change in MEP amplitude for the three post-kTMP blocks for the 3.5 kHz condition (mean ± standard error).

Limiting our analysis to the four active AM conditions, we found that MEP amplitude was not dependent on modulating frequency [$f_m$: $\chi^2(1) = 2.33$, p = 0.127] or burst repetition frequency [$f_b$: $\chi^2(3) = 3.92$, p = 0.27]. We found an effect of Time (i.e., Post Block) on MEP amplitude across all AM conditions [$\chi^2(2) = 7.45$, p = 0.024], but no significant interaction between Post Block and Stimulation condition [$\chi^2(6) = 3.14$, p = 0.791]. Pairwise comparisons showed that MEP amplitude was greater in Post Block 3 compared to Post Block 1 [$\chi^2(1) = 7.75$, p = 0.005; *Figure 4*: AM conditions]. No other comparisons were significant [all $\chi^2$'s < 2.30, all p's > 0.129].

Our main interest in the AM conditions is to determine if amplitude modulation at physiologically relevant frequencies produces a change in cortical excitability beyond that produced by the kHz carrier frequency. To this end, we compared the effect of AM kTMP to non-modulated kTMP, with the latter limited to the 3.5 kHz condition (pooled data). Using a mixed-effects model, this contrast was not significant [$\chi^2(1) = 0.05$, p = 0.833; $d = 0.03$]. Thus, the AM conditions provide another demonstration that kHz stimulation can increase cortical excitability. However, at least with a 2 V/m E-field, we did not observe an effect from the AM component that was above and beyond non-modulated kTMP.

## Paired-pulse assays

We included two paired-pulse assays that provide probes of short intracortical inhibition (SICI) or facilitation (ICF). To confirm that we observed an effect from the paired-pulse protocols, we focused on the pre-kTMP data, thus avoiding any influence from kTMP stimulation. We pooled the data across the two pre-kTMP blocks for each individual and performed an analysis on the data from the participants

from Experiments 1 and 2. This analysis showed a strong SICI effect, with a mean decrease of 36% on the paired-pulse trials relative to the SP trials [$t(125) = -24.47$, $p < 0.001$; **Figure 5**, top]. In addition, we saw a facilitatory effect for the ICF protocol with a mean increase of 22% on the paired-pulse trials relative to the SP trials [$t(125) = 6.86$, $p < 0.001$; **Figure 5**, bottom].

## No effect of kTMP on measures of intracortical inhibition or facilitation

We then asked if the magnitude of the SICI effect was modulated by kTMP (**Figure 5**, top). Compared to sham, we observed no effect of active non-modulated kTMP [$\chi^2(1) = 0.81$, $p = 0.367$], no effect of Time (i.e., Post Block) [$\chi^2(2) = 0.53$, $p = 0.768$] and no interaction between these two factors [$\chi^2(6) = 5.47$, $p = 0.486$]. Similarly, there was no effect of active AM kTMP on SICI (relative to sham) [$\chi^2(1) = 0.56$, $p = 0.455$], no effect of Time [$\chi^2(2) = 0.39$, $p = 0.822$] and no interaction between the two factors [$\chi^2(4) = 4.17$, $p = 0.384$]. In sum, kTMP does not appear to influence the magnitude of intracortical inhibition measured by SICI.

As with SICI, there was no evidence that kTMP influenced ICF (**Figure 5**, bottom): We found no effect of active non-modulated kTMP compared to sham [$\chi^2(1) = 0.318$, $p = 0.572$], no effect of Time (i.e., Post Block) [$\chi^2(2) = 1.48$, $p = 0.478$], and no interaction between the two factors [$\chi^2(6) = 5.07$, $p = 0.534$]. Similarly, we found no effect of active AM kTMP compared to sham [$\chi^2(1) = 0.11$, $p = 0.745$],

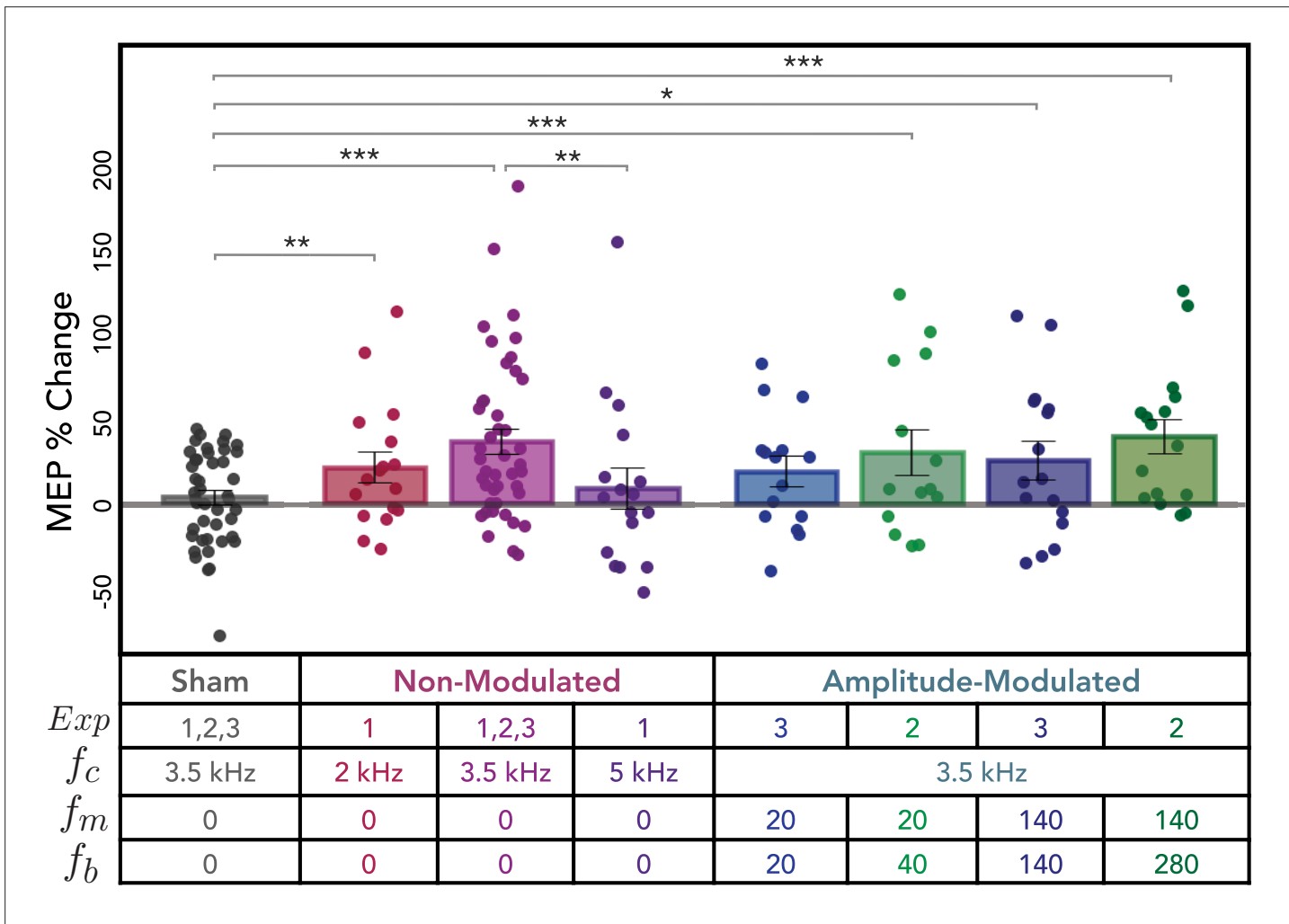

| | Sham | Non-Modulated | | | Amplitude-Modulated | | | |
|---|---|---|---|---|---|---|---|---|
| $Exp$ | 1,2,3 | 1 | 1,2,3 | 1 | 3 | 2 | 3 | 2 |
| $f_c$ | 3.5 kHz | 2 kHz | 3.5 kHz | 5 kHz | 3.5 kHz | | | |
| $f_m$ | 0 | 0 | 0 | 0 | 20 | 20 | 140 | 140 |
| $f_b$ | 0 | 0 | 0 | 0 | 20 | 40 | 140 | 280 |

**Figure 3.** Post-stimulation changes in cortical excitability as measured by single-pulse transcranial magnetic stimulation (TMS) for all conditions. Percent change in motor-evoked potential (MEP) amplitude following sham and active kHz stimulation, relative to baseline. Dots denote values for individual subjects, bars—mean values, and whiskers—standard error. Note that the data for the sham and non-modulated 3.5 kHz conditions are combined across the three experiments. *p < 0.05, **p < 0.01, ***p < 0.001.

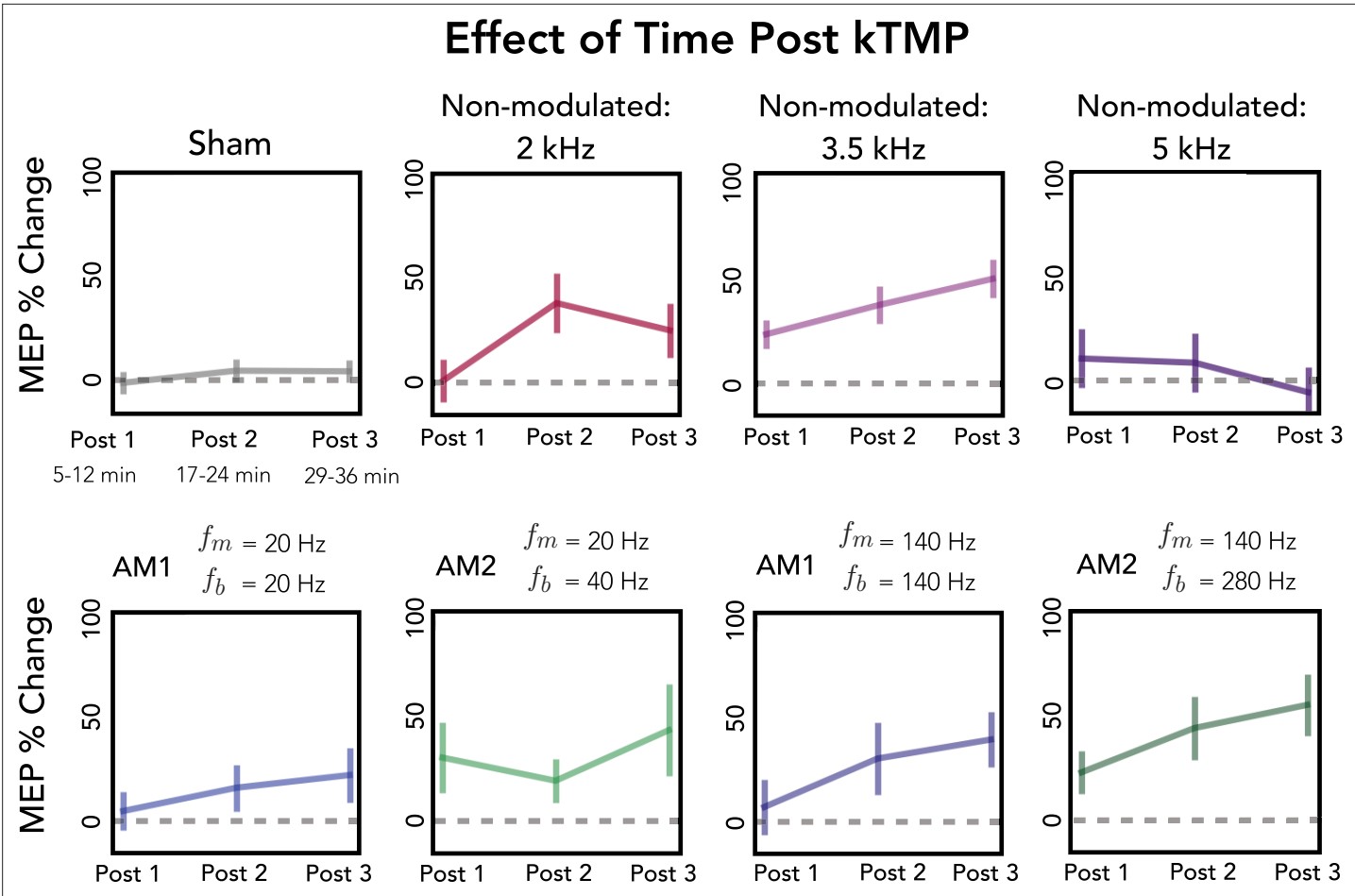

**Figure 4.** Post-stimulation change in cortical excitability as measured by single-pulse transcranial magnetic stimulation (TMS) across the three post-kilohertz transcranial magnetic perturbation (kTMP) blocks. Percent change in motor-evoked potential (MEP) amplitude following sham and active kHz stimulation, relative to baseline for the three post blocks (error bars represent standard error). The data for the sham and non-modulated 3.5 kHz conditions are combined across the three experiments.

no effect of Time [$\chi^2(2) = 0.71$, p = 0.702] and no interaction between the two factors [$\chi^2(4) = 2.43$, p = 0.657].

In summary, kTMP, either non-modulated or amplitude modulated at 2 V/m, did not produce any measurable effect on the paired-pulse assays of intracortical inhibition or facilitation, SICI and ICF.

### Subjective experience during kTMP stimulation

Informal observations from the participants in Experiment 1 indicated that the coil did not produce any detectable tactile sensation. The amplifier produces a sound, but one that was effectively masked by playing a louder, pre-recorded sound (3.5 kHz tone) in all conditions.

We opted not to ask participants to judge if they were in an active or sham condition. Given that each participant returned for multiple sessions, we did not want to alert them to the presence of a sham condition, preferring to simply describe the study as one testing a new method of NIBS. As such, we focused on their subjective ratings. Participants provided three ratings using an 11-point scale (0 = not at all; 10 = extremely) in response to questions on annoyance, pain, and subjective experience of muscle activation/movement. *Figure 6* presents the data for each measure following TMS, active kTMP, and sham kTMP. In line with expectations for TMS (*Meteyard and Holmes, 2018*), the participants were aware of muscle twitches, but the stimulation was well tolerated in terms of annoyance and pain given that the coil was positioned over primary motor cortex.

Of primary interest for the present report, the modal rating was 0 for all three ratings following active kTMP (*Figure 6*). Using a mixed-effects model, with a fixed factor of Stimulation type (i.e.,

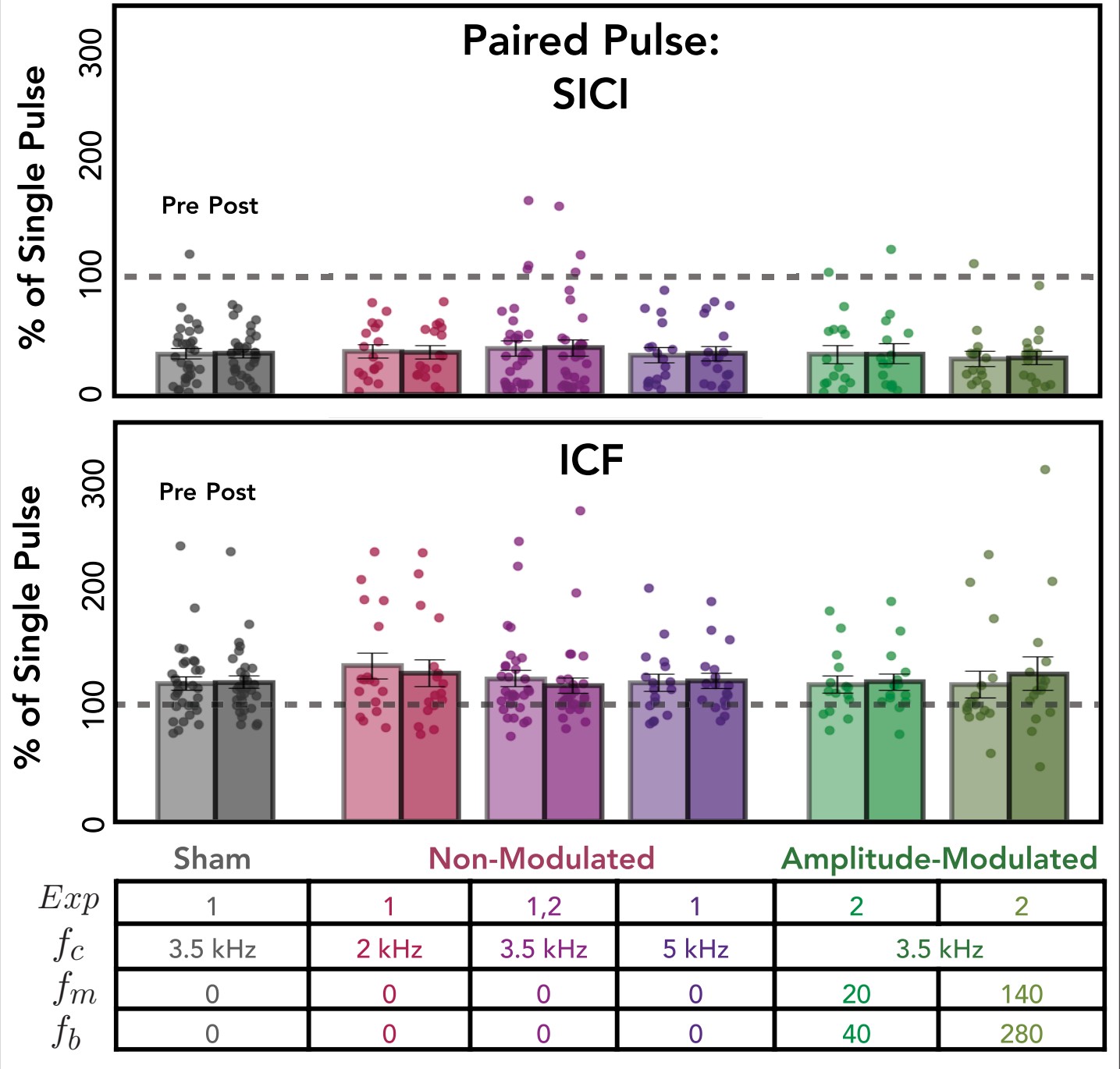

**Figure 5.** Kilohertz transcranial magnetic perturbation (kTMP) did not produce any change in measures of short intracortical inhibition (SICI) or intracortical facilitation (ICF). Data are plotted as the ratio of the paired-pulse motor-evoked potential (MEP) amplitude over the single-pulse MEP amplitude, with an inter-pulse interval of 3 ms for SICI and 10 ms for ICF. Each pair of bars shows this ratio for pre-kTMP (averaged over two probe blocks) and post-kTMP (averaged over three probe blocks).

active vs. sham) and Participant as a random effect, we found no difference between active and sham kTMP for Annoyance [$\chi^2(1) = 0.18$, $p = 0.672$], Pain [$\chi^2(1) = 0.29$, $p = 0.591$] and the subjective experience of Muscle Twitches [$\chi^2(1) = 0.01$, $p = 0.972$; *Figure 6*]. Some participants rated active and sham kTMP high in terms of annoyance. Based on open-ended survey responses, these scores appear to be related to general features of the experiment such as tension from the neuronavigation headband and

the requirement to maintain a sitting posture for an extended period. In summary, the survey data and subjective reports suggest that kTMP is well suited for double-blind experimentation.

## Discussion

In this paper, we present a new method, kTMP, for exploring the subthreshold NIBS experimental space. The kTMP E-field has the waveform flexibility of kilohertz tES along with the potential amplitude range and focality of TMS. Moreover, amplitude modulation of the kTMP E-field yields a waveform that could potentially introduce stimulation dynamics at frequencies matching endogenous neural rhythms (e.g., alpha and beta).

### Summary of experimental findings and limitations

Across three experiments, we assessed the ability of non-modulated and AM kTMP to alter cortical excitability, using suprathreshold TMS over motor cortex to elicit MEPs in a finger muscle. In Experiment 1, non-modulated kTMP with a targeted cortical E-field of 2 V/m produced an increase in corticospinal excitability at 2 and 3.5 kHz. The latter condition also resulted in a larger increase in MEP amplitude compared to stimulation at 5 kHz, suggesting a possible effect of carrier frequency. Experiments 2 and 3 provided replications of the efficacy of non-modulated kTMP at 3.5 kHz to increase corticospinal excitability. Importantly, participants experienced no tactile sensation or discomfort from the procedure. Indeed, the only percept of kTMP at 2.0 V/m is a tone at the carrier frequency emanating from the amplifier. This sound was masked in our experiments by a background tone played during active and sham kTMP stimulation.

Due to pandemic-related university regulations, we were limited to a 2-hr experimental session. As such, we were only able to assess the impact of kTMP on neural excitability in three probe blocks, spanning a 36-min post-stimulation epoch. We did not observe a reduction of the kTMP effect across this window. Future testing with extended post-stimulation assays will be required to establish the duration of the effect of kTMP on cortical excitability.

We also tested two forms of AM, using modulation frequencies of 20 and 140 Hz. An increase in corticospinal excitability was observed in three of the four conditions, associated with high burst repetition frequencies. Critically, none of the AM kTMP conditions produced an increase above that observed with non-modulated 3.5 kHz kTMP, suggesting that at least for an E-field amplitude of

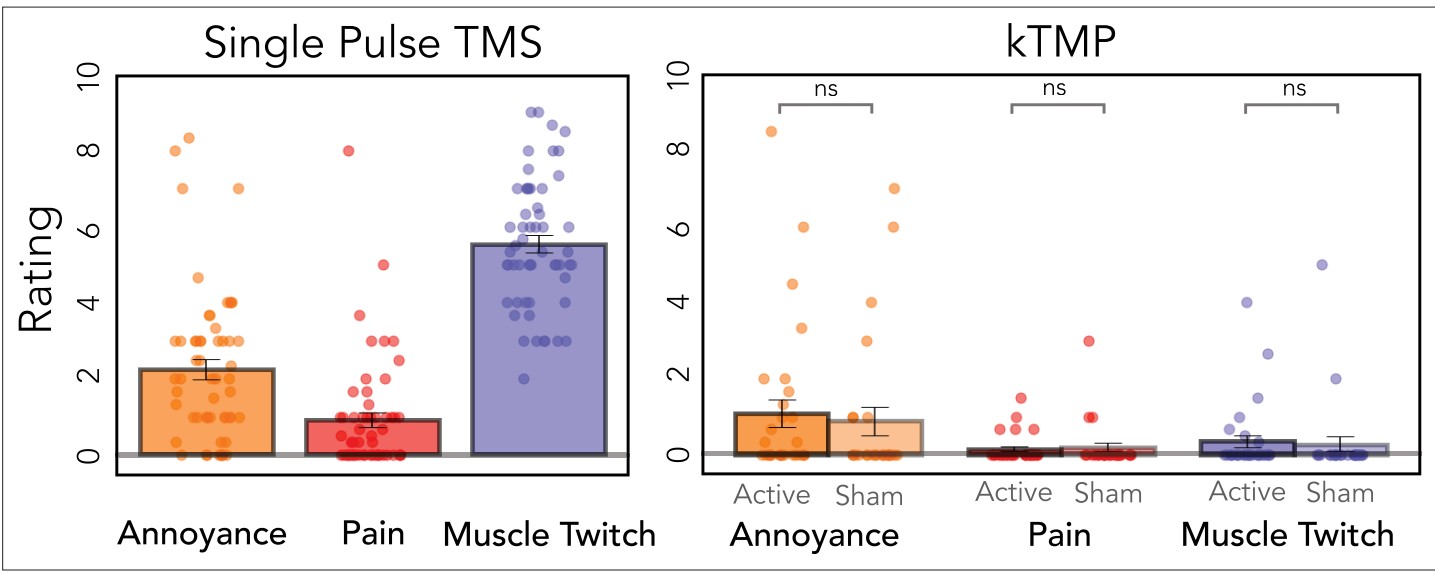

**Figure 6.** Kilohertz transcranial magnetic perturbation (kTMP) stimulation is not associated with any subjective experience. Mean ratings (SE in parenthesis) combined across Experiments 2 and 3 on a 0-to-10 scale in response to questions assessing annoyance, pain, and awareness of finger movement. For transcranial magnetic stimulation (TMS), the survey was administered after the second baseline probe block; for kTMP, the survey was administered after kTMP stimulation.

2 V/m, amplitude modulation at physiologically relevant frequencies does not induce an additional lasting change in corticospinal excitability over that arising from the kHz carrier frequency alone.

Despite the absence of an effect from amplitude modulation, the present results are encouraging in that, across a range of conditions, we consistently observed an increase in cortical excitability in response to SP TMS following the application of a 2 V/m E-field for 10 min. On average, the increase in MEP amplitude following 3.5 kHz non-modulated kTMP with a cortical E-field of 2 V/m, the condition in which we have the largest sample size, was 37%. This value is in the range observed from studies using a variety of NIBS methods and protocols (*Chaieb et al., 2011*; *Wischnewski et al., 2019*; *Bastani and Jaberzadeh, 2013*).

## Potential mechanisms of kTMP

At present, we can only tentatively speculate about how kTMP modulates cortical excitability. As with other NIBS modalities, a mechanistic understanding would address (1) what cellular element is directly perturbed, (2) which cortical neurons are modulated, and (3) how modulation confers lasting cortical excitability. We discuss these questions below.

Regarding acute effects, suprathreshold TMS activates neurons by inducing a brief E-field pulse with dominant frequency component in the kilohertz range (2–6 kHz) (*Peterchev et al., 2021*), demonstrating that such frequencies can acutely affect neuronal polarization. While the slower time constants of the soma and dendrite membranes mostly filter out kilohertz frequencies, the faster time constants of the axonal membrane allow kilohertz waveforms to produce transient polarization in the axons, especially axonal (presynaptic) terminals (*Aberra et al., 2020*; *Aberra et al., 2018*; *Barker et al., 1991*; *Nowak and Bullier, 1998a*; *Nowak and Bullier, 1998b*). Whereas suprathreshold TMS pulses induce action potentials in the axons, kTMP would only produce subthreshold perturbation of the axonal membrane potential. Thus, the effects of kTMP may be mediated by acute subthreshold polarization perturbation of presynaptic terminals.

Regarding the modulation of specific cortical circuits, the single- and paired-pulse TMS measurements can provide some insights; it has to be recognized, however, that these results depend on both the intervention (kTMP) as well as the TMS probe characteristics. The shape of the TMS pulse and coil orientation impact how a suprathreshold TMS pulse elicits an MEP: The MEP can come about via direct excitation of the axons of the corticospinal neurons (D-wave), via indirect effects on neurons that provide synaptic input to corticospinal neurons (I-waves), or a combination of both. We applied the TMS probes using suprathreshold biphasic TMS pulses with a posterior–anterior dominant induced current. At the probing stimulation level (120% resting motor threshold [rMT]), this configuration has been shown to preferentially recruit I-waves (*Di Lazzaro et al., 2001*). Moreover, since kTMP used the same coil configuration, it can be inferred that the kTMP E-field coupled most strongly to the same neural elements as the TMS probing pulses. Assuming that we are unable to probe the impact of kTMP on D-waves, we can speculate that the plasticity induced by kTMP predominantly affects excitatory synaptic inputs to pyramidal tract neurons. This mechanistic hypothesis is similar to that proposed for intermittent theta burst stimulation (iTBS), another subthreshold magnetic induction protocol that produces a temporally extended increase in SP MEPs (*Di Lazzaro and Rothwell, 2014*). However, it will be important to employ alternative coil configurations in future research to assess if kTMP can modulate the excitability of D-waves, representing direct activation of the axons of pyramidal tract neurons.

We did not observe any effect of kTMP on the two paired-pulse protocols, SICI and ICF. They appear to probe, respectively, GABA-mediated inhibitory and glutamate-mediated excitatory cortical circuits (*Cash and Ziemann, 2021*). Although we observed the classic signatures of SICI and ICF, kTMP did not modulate either of these measures of local intracortical neural dynamics. This pattern matches that observed with iTBS: A meta-analysis of iTBS studies failed to find consistent changes in SICI or ICF despite a consistent increase in SP MEPs (*Chung et al., 2016*). In contrast, a meta-analysis of anodal tDCS revealed an increase in SP MEP amplitude, a decrease in SICI, and an increase in ICF (*Biabani et al., 2018a*). Interestingly, tACS at 20 Hz also produced a similar pattern on all three measures in a phase-dependent manner during stimulation (*Guerra et al., 2016*). It is notable that kTMP and iTBS share the same spatial distribution of the E-field (determined by the TMS coil configuration), which is very different from that of the electrical stimulation with tDCS and tACS. Moreover, both TMS and kTMP involve stimuli in the kHz spectrum whereas tDCS and 20 Hz tACS use E-fields that

vary on slower time scales. This variation can result in a different pattern of polarization across neural elements. The overlapping characteristics shared by kTMP and iTBS may account for the similarity in coupling to specific cortical circuits which differ from those affected by tDCS and low-frequency tACS.

Finally, we can consider different hypotheses concerning how kHz stimulation produces changes in neuronal activity lasting beyond the stimulus train. One possibility is that the continuous acute perturbation of the presynaptic terminals interacts with the endogenous axonal signaling, for example via stochastic resonance, ultimately causing synaptic potentiation (*Antal and Paulus, 2013*). Another option is that the nonlinear properties of the neuronal membrane lead to temporal summation of successive kilohertz subthreshold cycles and, consequently, a facilitation of synaptic transmission (*Neudorfer et al., 2021*). The repeated perturbation of the neuronal membrane could also result in accumulation of calcium in presynaptic terminals, yielding synaptic plasticity effects (*Chaieb et al., 2011*). Presently, these possibilities are speculative and need to be assessed by further research into the cellular mechanisms of subthreshold kilohertz neuromodulation.

## Methodological advantages of kTMP

A limiting factor for tES studies is that these methods are restricted to the very low end of subthreshold space in terms of E-field amplitude. Thus, there is a limited range over which one could obtain dose-dependent effects; indeed, efforts to establish dose-dependent response functions with NIBS protocols have provided mixed results (*Kamen, 2004*; *Malcolm et al., 2006*). In contrast, kTMP has the potential to open a large subthreshold experimental space, one that has considerable range in terms of E-field amplitude, carrier frequency, AM waveform, and stimulation duration. We should be able to achieve at least a fourfold increase in the kTMP cortical E-field amplitude (e.g., up to 8 V/m in the kHz range) (*Sheltraw et al., 2021*; *Wang et al., 2023*) without uncomfortable scalp stimulation. This expanded parameter space, especially in terms of E-field amplitude, should prove beneficial in deriving dose-dependent response functions. Theoretically, kTMP could reach suprathreshold E-field amplitudes, although moving into that space will require careful assessment of issues related to power requirements, coil heating, acoustic noise, and participant safety.

In addition to offering a large subthreshold experimental space, there are other noteworthy features of kTMP. First, kTMP is ideally suited for double-blind experimentation. As verified in the subjective reports of our participants, the only percept associated with 2.0 V/m kTMP is a tone at the carrier frequency emanating from the amplifier, one that can be masked. Informally, we have placed the coil at different locations on the scalp and found that, even when positioned over inferior prefrontal or occipital cortex, there is no scalp stimulation or tactile percept from kTMP stimulation, issues that can impact TMS and tES protocols (*Meteyard and Holmes, 2018*; *Matsumoto and Ugawa, 2017*).

Second, for studies using TMS as a probe of NIBS efficacy, the E-fields of the perturbation (e.g., kTMP) and probe (e.g., suprathreshold SP TMS) have matched spatial distribution and only differ in terms of their waveforms and strengths. In contrast, the E-fields of tES and TMS cannot be matched (*Sheltraw et al., 2021*) and, as such, likely impact different neural populations even when the targeted region is ostensibly the same. Beyond the experimental convenience of using the same TMS coil for both perturbation and probe as in our prototype, the E-field alignment may increase experimental robustness.

Third, the experimenter can create a kTMP waveform of unlimited flexibility. For example, using AM kTMP to introduce perturbations at physiologically relevant frequencies could provide a method to enhance plasticity or even induce neural entrainment. Although we failed to observe an additional effect of AM kTMP, research using non-human models suggests that entrainment effects from NIBS require strong E-fields since the exogenous stimulation pattern must compete with endogenous brain rhythms (*Liu et al., 2018*; *Wang et al., 2023*; *Khanna et al., 2021*). As such, kTMP offers an approach that combines an expanded range of subthreshold E-field amplitudes with waveform flexibility in the kHz range.

Fourth, E-fields produced through magnetic induction, unlike tES E-fields, depend weakly upon the distribution of tissue conductivity—estimates of which vary widely in the literature (*Hallez et al., 2007*). Indeed, for spherical shell head models, the E-field produced through magnetic induction is independent of conductivity—something that does not hold for tES E-fields. This should reduce variability when modeling the E-field based on individualized head geometry.

Fifth, as with other NIBS methods (*Ghafoor et al., 2022*; *Herrmann et al., 2016*; *Fehér and Morishima, 2016*; *Di Gregorio et al., 2022*), we anticipate that simultaneous EEG and kTMP will enable the investigation of the evolution of neural effects in real-time. For example, with amplitude modulated kTMP, the artifact from the kHz carrier frequency can be removed, albeit with possible technical hurdles connected to nonlinearities associated with presently available recording and stimulation hardware (*Kasten et al., 2018*). This approach could be used to tailor the stimulation waveform on an individual basis or for closed-loop control, promising avenues for translational applications of NIBS.

## Conclusion

In conclusion, kTMP offers an opportunity to explore a new experimental space, one with a relatively large range of subthreshold E-field induction, the focality of TMS, and the potential to impose E-fields at physiologically relevant frequencies, with significantly less tolerability issues than tES or suprathreshold TMS.

## Methods

### Apparatus

The kTMP system consists of a high-amplitude current source, a TMS coil, and a control system. The same TMS coil may be connected to either the kTMP current source or to a commercial TMS pulse generator (MagVenture MagPro R30 with MagOption), permitting interleaved kTMP–TMS experiments and ensuring identical kTMP and TMS E-field distributions up to an amplitude scaling factor. The kTMP amplifier (AE Techron Model 7794) is a voltage-controlled current source capable of delivering up to 200 A to the coil. We used an actively liquid-cooled figure-of-eight coil (MagVenture Cool-B65; inner and outer loop diameter of 35 and 75 mm, respectively).

The kTMP control system consists of a personal computer (PC), input/output PCI card, and a custom interface to read the coil's built-in temperature sensor. Using a data acquisition toolbox (Mathworks R2018a), the PCI card was programmed to deliver analog input to the amplifier, thus specifying the temporal waveform of the E-field. The input waveform can either be at a fixed-amplitude sinusoidal frequency (non-modulated) or amplitude modulated.

Bench testing indicated that the system when running in kTMP mode did not produce marked changes in coil temperature. As an added safeguard, the PCI card was set up to receive an analog input from the coil's temperature sensor and create an automatic shutdown if the coil temperature exceeded 41°C, which is within the guidelines established by the International Electrotechnical Commission (IEC). In practice, for E-fields up to 2 V/m used in the present experiments, the coil temperature never rose above 32°C during system operation.

### Participants

Forty-nine young adults were recruited through various advertisements posted to the Berkeley community. The number of participants in Experiments 1, 2, and 3 was 16 (12 female), 16 (10 female), and 15 (11 female), respectively. Seven of the participants in Experiment 2 had also been tested in Experiment 1 (minimum 3 weeks between the last session of Experiment 1 and first session of Experiment 2). All participants provided informed consent, were compensated for their time, and were naïve to the purpose of the study. Eligibility was determined in accordance with current TMS safety guidelines (*Rossi et al., 2021*), and individuals with contraindications to TMS were excluded. Given the novelty of kTMP as a NIBS method, the UC Berkeley Institutional Review Board (IRB) consulted with an outside expert and members of the campus Environmental and Health Safety Committee to evaluate the system. Following their review, the protocol was approved by the UC Berkeley IRB (Protocol Number 2018-05-11093), with the kTMP system classified as a non-significant risk device for E-fields up to 8 V/m applied for up to 20 min per session.

### Procedure

To evaluate the kTMP system as a new tool to modulate neuronal excitability, we measured the impact of kTMP on corticospinal excitability using suprathreshold TMS stimulation over motor cortex. *Figure 7* depicts an overview of the experimental hardware and protocol. In brief, kTMP stimulation

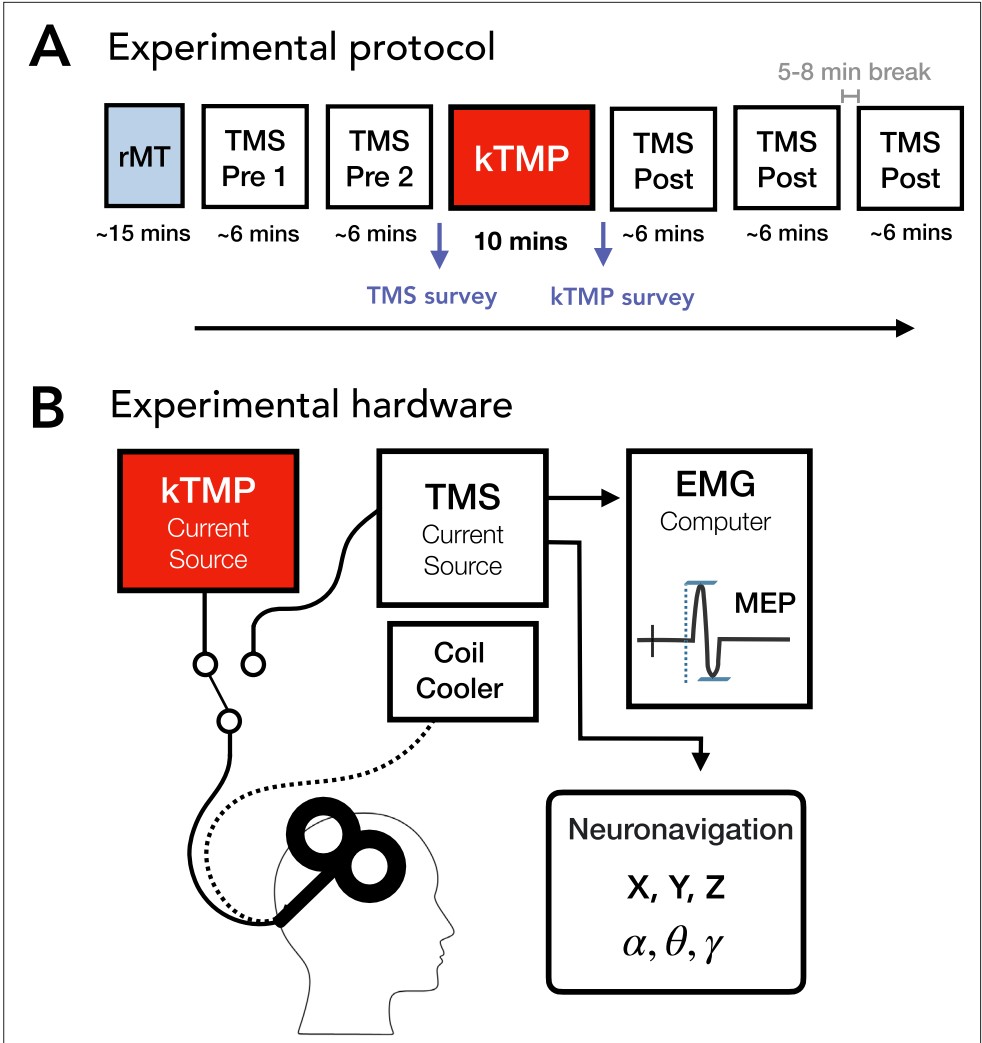

**Figure 7.** Experimental protocol and hardware. (**A**) Timing of each experimental session. After determining the participant's resting motor threshold (rMT), TMS assessment blocks were conducted before (Pre) and after (Post) kTMP stimulation (active or sham). For Experiments 1 and 2, each TMS block assessed single-pulse, short intracortical inhibition (SICI), and intracortical facilitation (ICF); only the single-pulse protocol was used in Experiment 3. (**B**) The same coil was driven by either the TMS current source or the kTMP current source. TMS pulses were recorded in an auxiliary channel of the electromyography (EMG) and triggered the neuronavigation system to record the coordinates of the coil in 3D space.

was preceded by two 4–7 min probe blocks and followed by three such blocks. In Experiments 1 and 2, each block consisted of SP TMS and two paired-pulse protocols, SICI and ICF. In Experiment 3, only the SP TMS protocol was employed (see below for details).

Each experiment consisted of five 2 hr test sessions, with each session separated by a minimum of 2 days. The first test session was used to determine the optimal coil position (hotspot) and threshold intensity for eliciting MEPs with suprathreshold SP TMS (see below). The position of the hotspot was recorded by a neuronavigation system (Brainsight, Rogue Research, Montreal, Canada). This allowed the experimenter to return to the same position for each TMS block (see below), as well as during the application of kTMP. The other four sessions were used to test the efficacy of different kTMP parameters on cortical excitability, with a focus on variation of the carrier frequency for non-modulated kTMP in Experiment 1 and both non-modulated and AM kTMP in Experiments 2 and 3.

Two steps were taken to create a double-blinding protocol. First, we created a coding system such that the experimenter typed in a number that was paired to the desired stimulation condition in an arbitrary and random manner, one that varied across participants in a manner unknown to

the experimenter. Second, we played a tone at 3.5 kHz using the Tone Generator Application Sonic (VonBruno 2015) to create a constant background sound during kTMP stimulation in all conditions, including sham, effectively masking the amplifier sound.

## kTMP

For each carrier frequency, the current amplitude was selected to achieve a peak E-field amplitude of 2.0 V/m at a distance of 14 mm perpendicular to the coil surface, a distance we took to represent the approximate depth of the motor cortex from the overlying scalp (*Lu et al., 2019*). The current amplitude needed to obtain the required E-field amplitude is estimated as follows. The E-field amplitude, $E_0$ (units of V/m), is proportional to the frequency, $f_b$ (units of Hz), and amplitude $I$ (units of A) of the current source according to:

$$E_0 = k f_c I.$$

For the MagVenture Cool-B65 coil used in the kTMP system $E_0$ = 185 V/m when $f_c$ = 3570 Hz (280 µs biphasic TMS pulse) and $I$ = 6,950 A (*Deng et al., 2013*; *Drakaki et al., 2022*). From these reference values, we estimate $k$:

$$k = 7.875 \times 10^{-6} \frac{\text{Vs}}{\text{Am}}.$$

where V, s, A, and m correspond to volts, seconds, amperes, and meters, respectively.

We verified the accuracy of our estimates within a 5% error using E-field measurements obtained from a triangular probe following the method of Nieminen (*Nieminen et al., 2015*).

## Experiment 1
### Non-modulated kTMP

We used a within-subject design, testing each participant on each of four kTMP stimulation conditions, with the order counter-balanced across participants. For three of the conditions, the carrier frequency (2, 3.5, and 5 kHz) was paired with an intensity to create an E-field at the superficial aspect of the hand area of the motor cortex of $E_0$ = 2 V/m. We set the non-modulated E-field to be a sine wave with frequency $f_c$ (see *Figure 8*):

$$E_{NM}(t) = E_o \cos(2\pi f_c t).$$

Note that we did not adjust for individual differences in scalp-to-cortex distance. For the sham condition, we used a 3.5-kHz carrier frequency producing a 0.01 V/m E-field at the approximate distance of the cortical surface.

## Experiments 2 and 3
### Amplitude-modulated kTMP

For Experiments 2 and 3, we again used a within-subject design, testing each participant in four sessions. Two of the conditions were repeated from Experiment 1: the 3.5 kHz non-modulated kTMP condition at 2 V/m and the sham condition. For the other two conditions, the carrier frequency was set at 3.5 kHz and the waveform was amplitude modulated (AM) at modulation frequencies of either 20 or 140 Hz. The 3.5 kHz carrier frequency was chosen since we had obtained the largest effect size compared to sham at this frequency in Experiment 1. We selected the 20 Hz modulation frequency given the relevance of beta to motor function (*Feurra et al., 2013*; *Feurra et al., 2011*; *Heise et al., 2016*) and the 140 Hz modulation frequency based on literature concerning ripple effects at this frequency (*Dissanayaka et al., 2017*; *Inukai et al., 2016*; *Moliadze et al., 2010*). The peak cortical E-field amplitude for the AM conditions was 2 V/m, identical to the non-modulated condition. Note that the inclusion of the 3.5 kHz non-modulated condition and sham for both experiments provides two replications of these conditions from Experiment 1 as well as points of comparison for the AM kTMP conditions.

Experiments 2 and 3 differed in the form used for amplitude modulation. In general, an amplitude-modulated current can be written as

$$E(t) = A(t) \sin(2\pi f_c t),$$

where $A(t)$ is a time-dependent amplitude modulation factor. A popular form used in communications systems is

$$A(t) = E_o [a + mf(t)],$$

where $f(t)$ is the signal one wishes to convey between two components of a system and the ratio $\frac{m}{a}$ is the modulation index. The modulation index is typically chosen to suit the proposed means of demodulation.

In *Experiment 2*, we used $f(t) = \sin(2\pi f_m t)$ and an infinite modulation index. Correspondingly, the E-field has time dependence of the form:

$$E_{AM1}(t) = E_o \sin(2\pi f_m t) \sin(2\pi f_c t),$$

where $f_m$ is the modulation frequency. In Experiment 3, we used $f(t) = \sin(2\pi f_m t)$ and a modulation index of one (see *Figure 8*),

$$E_{AM2}(t) = E_o \left[\frac{1}{2} + \frac{1}{2}\sin(2\pi f_m t)\right]\sin(2\pi f_c t).$$

Although the same modulation frequencies are used to calculate $E_{AM1}$ and $E_{AM2}$ they differ with respect to the spectrum of their upper envelope, the frequency of which we refer to as the burst repetition frequency $f_b$. For $E_0$ the burst repetition frequency is double that of the modulation frequency, whereas for $E_{AM2}$ the burst repetition frequency and the modulation frequency are matched. The burst repetition frequency may be an important parameter in determining the neural effects of kTMP.

## TMS: hotspot and threshold procedure (Session 1)

SP TMS was applied over left hemisphere primary motor cortex to determine the rMT for the first dorsal interosseous (FDI) muscle in the right hand. We focused on FDI since it is relatively easy to isolate in all individuals and threshold values are stable across test sessions (*Kamen, 2004*; *Malcolm et al., 2006*; e.g., *Carroll et al., 2001*).

The TMS coil was placed tangentially on the scalp with the handle pointing backward and laterally at 45° from the midline. TMS was administered with a biphasic pulse waveform with a posterior–anterior direction of the second, dominant phase of the induced current. The stimulator intensity was initially set to 30% of maximal stimulator output and SPs were generated at 5-s intervals, with the experimenter visually monitoring the EMG output for MEPs. If no MEPs were detected after two or three pulses, the experimenter moved the coil a few mm. If a search over the candidate area failed to produce any MEPs, the stimulator output was increased (step size of 3%), with the location search repeated. Once MEPs were detected, a more focal search was conducted to identify the optimal location for eliciting MEPs. This location was registered in three-dimensional space relative to the subject's head using the Brainsight neuronavigation system to ensure consistent coil position during and between experimental sessions. rMT was defined as the minimum TMS intensity required to evoke MEPs of at least 50 μV peak-to-peak amplitude on 5 of 10 consecutive trials.

The mean threshold was 58% (SD = 11.3%), 63% (SD = 11.1%), and 56% (SD = 9.2%) of maximum stimulator output in Experiments 1, 2, and 3, respectively. We repeated the threshold procedure in each of the kTMP sessions to capture possible intra-individual baseline changes in the cortico-excitability of the participants. In practice, the individual's threshold values remained very stable across days (SD = 2.4%).

## TMS assays of corticospinal excitability (Sessions 2–5)

For Experiments 1 and 2, each of the five probe blocks (two pre-kTMP and three post-kTMP) included SP TMS, and two paired-pulse protocols: SICI and ICF (*Cash and Ziemann, 2021*; *Rossini et al., 2015*). Similar to SP, paired-pulse protocols were administered with biphasic pulse waveforms with a posterior–anterior direction of the second, dominant phase of the induced current. These three proto-cols have been widely used in prior studies designed to evaluate the efficacy of tES and rTMS methods in altering neural excitability (*Chung et al., 2016*; *Horvath et al., 2015*; *Biabani et al., 2018a*). For

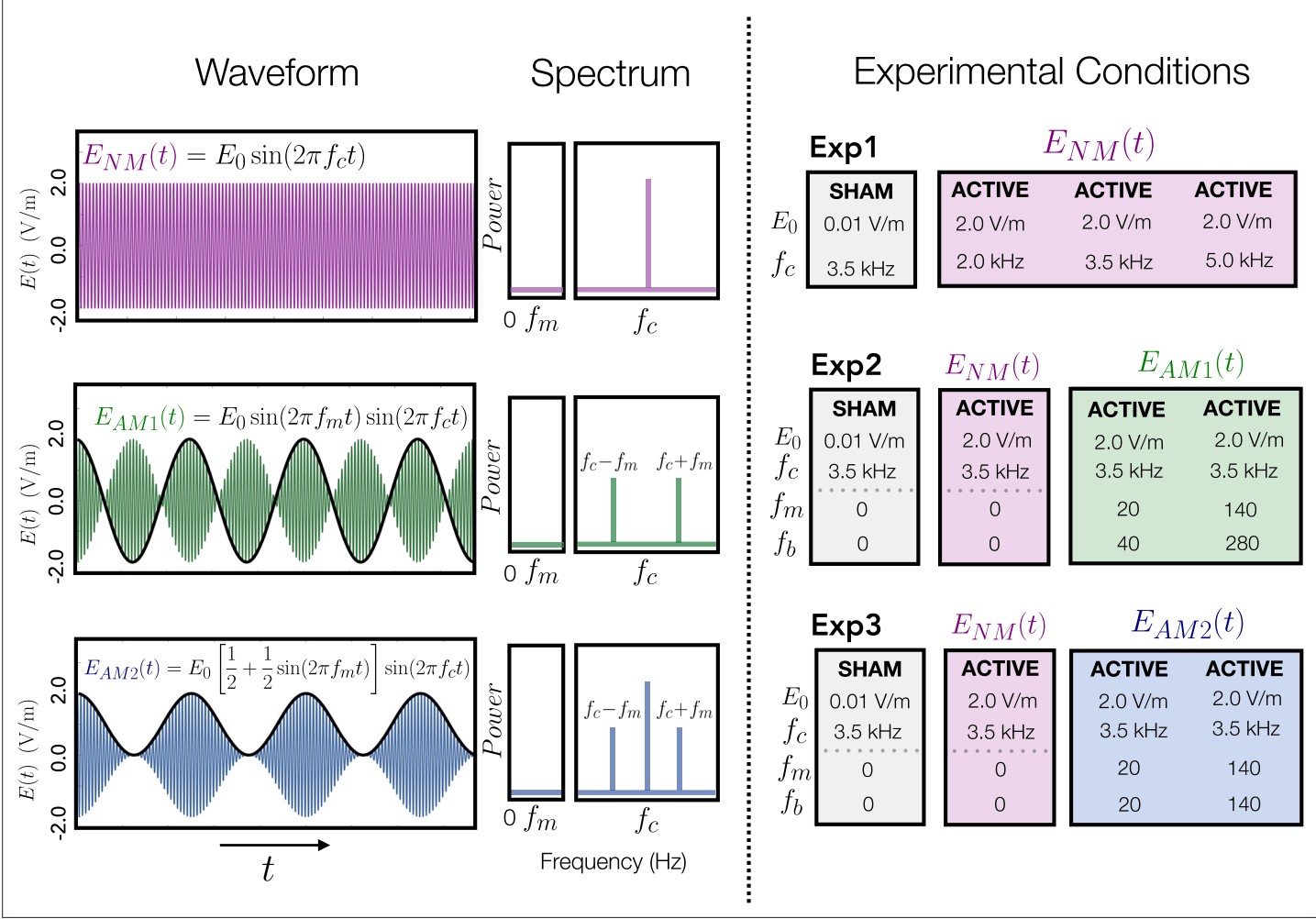

**Figure 8.** Waveforms, spectra, and conditions for the three kTMP experiments. A commonly used form of amplitude modulation is $E(t) = E_0 \left[ a + m \sin(2\pi f_m t) \right] \sin(2\pi f_c t)$. The constants $f_c, f_m$, and $E_0$ refer to the carrier frequency, modulation frequency, and the E-field amplitude (cortical E-field), respectively. Left column, waveforms: $E_{NM}, E_{AM1}$, and $E_{AM2}$ refer to the non-modulated waveform and the two forms of amplitude modulation tested. Black lines indicate the modulation frequency. Center column, spectrum: Carrier frequency and sidebands for the corresponding waveforms. Note the absence of power at the modulated frequency ($f_m$). Right column, waveform parameters and characteristics for each experiment. $f_c$ refers to the burst repetition frequency.

SP, the stimulation level was set at 120% of rMT. For the paired-pulse assays, the suprathreshold pulse was preceded by a subthreshold conditioning stimulus set at 80% of rMT, with an interstimulus interval (ISI) of 3 or 10 ms for SICI and ICF, respectively. The probe block consisted of 90 trials, 30 for each of the three assays, with the order randomized within the block. For Experiment 3, the five probe blocks consisted only of the SP TMS assay. We made this decision after finding that kTMP had no effect on SICI or ICF in Experiments 1 and 2. We lengthened the breaks between blocks in Experiment 3 to match the timing of the three post blocks relative to kTMP stimulation in the first two experiments.

We developed a system to read and record the spatial and angular position of the coil with respect to the hotspot in real time from Brainsight. This information was recorded at the time of each TMS pulse and used to exclude trials in which the coil was distant from the hotspot or the angle had changed from the optimal hotspot orientation.

## EMG

EMG activity was recorded (Bagnoli-8 EMG System, Delsys Inc) from surface electrodes placed over the right FDI muscles, with a reference electrode over the right elbow. The experimenter visually inspected the EMG traces on a monitor to ensure that the participant remained relaxed (i.e., negligible

EMG background activity in FDI), to detect the presence or absence of MEPs in response to the TMS pulses, and, since kTMP is a novel brain stimulation modality, to monitor for safety by checking for after discharges or other features suggesting excessive increase in excitability that could evolve into a seizure.

The EMG signal was amplified and bandpass filtered online between 20 and 450 Hz. The signals were digitized at 2000 Hz for offline analysis. All EMG measures were analyzed using custom MATLAB R2018a scripts, which are publicly available on GitHub (*Merrick et al., 2025*). EMG was recorded continuously during the experiment. Offline, data were segmented based on a TTL pulse from the TMS system recorded by the EMG amplifier on an auxiliary channel.

## Subjective reports

We informally assessed the participants' subjective experience in Experiment 1, focusing on reports concerning the perception of sound emitted from the system, tactile sensation, and discomfort. This process was formalized in Experiments 2 and 3. In these studies, we administered a short survey in which participants answered three questions concerning (1) annoyance, (2) pain, and (3) subjective experience of muscle activation/movement. Participants responded to each question using a keyboard to type in a number using an 11-point scale (0 = not at all; 10 = extremely). This survey was based on a systematic rating system that has been employed to characterize the degree of disturbance caused by TMS (*Meteyard and Holmes, 2018*). We administered the survey twice. The first time was after the second TMS probe block, providing ratings on the subjective experience of suprathreshold TMS. The second time was after kTMP, providing ratings on the subjective experience of kTMP stimulation.

## Data analysis

### MEP data

For each trial, the peak-to-peak amplitude of the MEP was calculated over a window 15–45 ms after the suprathreshold TMS pulse. Trials were excluded from the analysis based on the following criteria: (1) If the MEP amplitude was 2.5 standard deviations above or below the mean, with the mean and standard deviation calculated separately for each TMS assay (SP, ICF, and SICI) for each probe block. (2) If the Brainsight recording indicated that the coil was more than 3 mm (Euclidian distance) from the optimal hotspot location or had an angular or twist error more than 5° from the optimal trajectory angle. (3) If noise in the EMG signal 100 ms before the TMS pulse exceeded 2.5 standard deviations of the mean EMG signal. On average, 10% (SD = 3%) of the trials were excluded per participant with a range of 4.8–20%. After cleaning the MEP data, there were a minimum of 20 MEP measures per protocol in each assessment block for each individual, a sufficient number for performing the MEP analyses (*Chung et al., 2016*; *Goldsworthy et al., 2016*; *Cavaleri et al., 2017*; *Biabani et al., 2018b*).

Raw MEP amplitudes were log-transformed to normalize the distribution of MEP amplitudes (*Peterchev et al., 2013*; *Nielsen, 1996a*; *Nielsen, 1996b*; *Goetz et al., 2014*). The average log-transformed MEP amplitude was then calculated for each of the three TMS protocols in each probe block on an individual basis. After averages were calculated, the data were exponentiated to get the MEPs back to an easily interpreted scale (i.e., mV). SICI and ICF measures were calculated by computing a ratio of the paired-pulse MEP average over the SP MEP average for each block. For all three TMS assays, the effect of kTMP stimulation was operationalized as the average percent change post-kTMP relative to the two baseline blocks (averaged). For example, a value of 0% would indicate no change in SP MEP amplitude from pre- to post-stimulation, whereas a value of 100% would indicate the SP MEP amplitude doubled from pre- to post-stimulation. Thus, the main analysis focuses on the three post-kTMP stimulation probe blocks for each of the three TMS assays (SP, SICI, and ICF). All data are available on Dryad (*Merrick et al., 2025*) and the custom analysis scripts, including code to reproduce, are publicly available on GitHub (*Merrick et al., 2025*) under the MIT License, permitting free use, modification, and distribution.

### Missing data

Although we aimed for a fully within-subject design, a subset of the participants only had data for three of the four sessions. Missing data were due to (1) technical issues with the Brainsight neuronavigation system (*n* = 2/47), (2) university suspension of testing with human participants in March 2020 due to the onset of the COVID-19 pandemic (*n* = 5/47), or (3) determination that the results from a

session were a statistical outlier ($n = 3/47$). The latter three had an increase in corticospinal excitability three standard deviations above the mean in one condition. All of these were active kTMP conditions and, if included, would have inflated the effect of kTMP on corticospinal excitability. To account for missing data and subject variability, all analyses used a linear mixed-effects model with a random factor of participant.

### Linear mixed-effects models

Linear mixed-effects models were implemented in RStudio using the software package lme4 (*Bates et al., 2015*). Each mixed-effects model used Participant as a random effect, experiment as a fixed effect, and experimental variables (e.g., active vs. sham, post-stimulation block) as fixed effects. Likelihood ratio tests were used to obtain p-values in evaluating experimental fixed effects and interaction effects. Cohen's *d* was calculated based on the methods outlined in *Brysbaert and Stevens, 2018* for mixed-effects models. Cohen's *d* can be interpreted as a standardized mean difference between conditions; according to convention, $d \cong 0.2$ is considered a small effect, $d \cong 0.5$ is considered a medium effect, and $d > 0.8$ considered a large effect (*Cohen, 1988*).

## Acknowledgements

We are grateful to our team of research assistants, Tatianna Howard, Alice Wang, Serena Chi, Caroline Cao, Lauren Anne Schuck, Emily Schultz, Carolyn Irving, Lizzy Trinh, Kevin Peter, Owen Doyle, and Connor Brown. Assaf Breska provided advice on the statistical analyses. This work was supported by grants from the NIH (R21 EB028075; R35 NS116883; 1R44 NS127667) and NSF (EAGER 1946316) and intramural funding from the Henry H Wheeler, Jr. Brain Imaging Center. We thank MagVenture for providing the TMS stimulator and coil used in this study.

## Additional information

### Competing interests

Ludovica Labruna: LL has stock ownership of Magnetic Tides, a non-publicly traded company created to develop new methods of non-invasive brain stimulation. Christina Merrick: CM has stock ownership of Magnetic Tides, a non-publicly traded company created to develop new methods of non-invasive brain stimulation. Angel V Peterchev: AVP has stock ownership of Magnetic Tides, a non-publicly traded company created to develop new methods of non-invasive brain stimulation. Ben Inglis: BI has stock ownership of Magnetic Tides, a non-publicly traded company created to develop new methods of non-invasive brain stimulation. Richard B Ivry: RBI has stock ownership of Magnetic Tides, a non-publicly traded company created to develop new methods of non-invasive brain stimulation. Daniel Sheltraw: DS has stock ownership of Magnetic Tides, a non-publicly traded company created to develop new methods of non-invasive brain stimulation.

### Funding

| Funder | Grant reference number | Author |
| --- | --- | --- |
| National Institute of Biomedical Imaging and Bioengineering | R21 EB028075 | Ludovica Labruna<br>Ben Inglis<br>Richard B Ivry<br>Daniel Sheltraw |
| National Institute of Neurological Disorders and Stroke | R35 NS116883 | Richard B Ivry |
| National Institute of Neurological Disorders and Stroke | 1R44 NS127667 | Ludovica Labruna<br>Richard B Ivry<br>Daniel Sheltraw |

| Funder | Grant reference number | Author |
|---|---|---|
| National Science Foundation | EAGER 1946316 | Ludovica Labruna<br>Ben Inglis<br>Richard B Ivry<br>Daniel Sheltraw |

The funders had no role in study design, data collection, and interpretation, or the decision to submit the work for publication.

## Author contributions

Ludovica Labruna, Conceptualization, Data curation, Supervision, Funding acquisition, Investigation, Methodology, Writing – original draft, Project administration, Writing – review and editing; Christina Merrick, Data curation, Software, Formal analysis, Supervision, Validation, Visualization, Methodology, Writing – original draft, Project administration, Writing – review and editing; Angel V Peterchev, Methodology, Writing – review and editing; Ben Inglis, Conceptualization; Richard B Ivry, Conceptualization, Funding acquisition, Investigation, Project administration, Writing – review and editing; Daniel Sheltraw, Conceptualization, Funding acquisition, Project administration, Writing – review and editing

## Author ORCIDs

Ludovica Labruna ⬤ https://orcid.org/0000-0003-3978-591X
Christina Merrick ⬤ http://orcid.org/0000-0003-4966-9138
Angel V Peterchev ⬤ https://orcid.org/0000-0002-4385-065X
Richard B Ivry ⬤ https://orcid.org/0000-0003-4728-5130

## Ethics

All participants provided informed consent, were compensated for their time, and were naive to the purpose of the study. Eligibility was determined in accordance with current TMS safety guidelines, and individuals with contraindications to TMS were excluded. Given the novelty of kTMP as an NIBS method, the IRB at UC Berkeley enlisted an outside expert and members of the campus Environmental and Health Safety Committee to evaluate the system. Following their reports, the protocol was approved by the IRB at UC Berkeley (2018-05-11093) with the kTMP system deemed a non-significant risk device for E-fields up to 8 V/m for up to 20 min of stimulation.

Reviewer #1 (Public review): https://doi.org/10.7554/eLife.92088.3.sa1
Reviewer #2 (Public review): https://doi.org/10.7554/eLife.92088.3.sa2
Author response https://doi.org/10.7554/eLife.92088.3.sa3

---

# Additional files

## Supplementary files

MDAR checklist

## Data availability

All data supporting the findings of this study are openly available on Dryad at https://doi.org/10.5061/dryad.qz612jmsh. Analysis scripts for processing the data and code for generating Figures 4 and 5 are available on GitHub at (https://github.com/cmerrick15/kTMP_MEP_Analysis copy archived at *Merrick, 2025*).

The following dataset was generated:

| Author(s) | Year | Dataset title | Dataset URL | Database and Identifier |
|---|---|---|---|---|
| Merrick C | 2025 | Data for: Kilohertz transcranial magnetic perturbation (kTMP): A new non-invasive method to modulate cortical excitability | https://doi.org/10.5061/dryad.qz612jmsh | Dryad Digital Repository, 10.5061/dryad.qz612jmsh |

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
