## [Editor Report · eLife assessment]

This **important** study introduces and evaluates the efficacy of a novel form of non-invasive brain stimulation in humans: kilohertz transcranial magnetic perturbation (kTMP). The evidence provided for the ability of kTMP to increase cortical excitability with minimal sensation is **compelling**, with two separate replication experiments. Although exploratory in nature, this work represents new avenues for non-invasive brain stimulation research that has potential long-term appeal for both clinical and research applications. This paper will be of significant interest to neuroscientists interested in brain stimulation.

---

## [Referee Report · Reviewer #1 (Public review)]

Summary:

This paper reports the first results on the effects of a novel waveform for weak transcranial magnetic stimulation, which is refered to as "perturbation" (kTMP). The waveform is sinusoidal at kHz frequency with subthreshold intensities of 2V/m, instead of the suprathreshold pulses used in conventional TMS (~100V/m). The effect reported here concerns motor-evoked potentials (MEPs) elicited on the hand with single-pulse TMS. These MEPs are considered a marker of "corotico-spinal excitability". The manuscripts report that kTMP at 3.5kHz enhances MEPs with a medium effect size, with independent replication of this finding on 3 separate cohorts of subjects (N=16, 15, 16). This result is important for the field of non-invasive brain stimulation. The evidence in support of this claim is compelling. Despite the replications, this remains an exploratory study that will require replication with adequately powered planned comparisons.

Strengths:

• This is a novel modality for non-invasive brain stimulation.

• Knowing the history in this field, this is likely to lead to a large number of follow-up studies in basic and clinical research.

• The modality causes practically no sensation, which makes it perfectly suitable for control conditions. Indeed, the study itself used a persuasive double-blinding procedure.

• The replication of the main result in two subsequent experiments is very compelling.

• The effect size of Cohen's d=0.5 is very promising.

• It is nice the E-fields were measured on a phantom, in addition to modeling.

Weakness:

• Statistical analysis combining Experiments 1, 2, 3 after inspecting the data is inappropriate.

• Post-hoc definition of outliers that were removed is unfortunate.

• While sensation has been documented, blinding was not directly assessed.

• Despite the replications, this remains an exploratory study as it lacks power analysis and planned comparisons.

Other comments from an earlier review were adequately addressed.

---

## [Referee Report · Reviewer #2 (Public review)]

Summary:

kTMP is a novel method of stimulating the brain using electromagnetic fields. It has potential benefits over existing technology because it is a safe and easy technology. It explores a range of brain frequencies that has not been explored in depth before (2-5kHz) and thus offers new opportunities.

Strengths:

This work relied on standard methods and was carefully and conservatively performed.

Weaknesses:

There were few weaknesses. The sham condition was prepared as well as could be done, but sham is always challenging in a treatment with sound and sensation, and with knowledgeable operators. New technology, also, is very exciting to subjects and it is difficult to achieve a natural experiment. These difficulties are related to the technology, however, and not to the execution of these experiments..

---

## [Author Response]

The following is the authors’ response to the original reviews.

Response to Public Comments

(1) BioRxiv version history.

Reviewer 1 correctly noted that we have posted different versions of the paper on bioRxiv and that there were significant changes between the initial version and the one posted as part of the eLife preprint process. Here we provide a summary of that history.

We initially posted a bioRxiv preprint in November, 2021 (Version I) that included the results of two experiments. In Experiment 1, we compared conditions in which the stimulation frequency was at 2 kHz, 3.5 kHz, or 5.0 kHz. In Experiment 2, we replicated the 3.5 kHz condition of Experiment 1 and included two amplitude-modulated (AM) conditions, with a 3.5 kHz carrier signal modulated at 20 Hz or 140 Hz. Relative to the sham stimulation, non-modulated kTMP at 2 kHz and 3.5 kHz resulted in an increase in cortical excitability in Experiment 1. This effect was replicated in Experiment 2.

In the original posting, we reported that there was an additional boost in excitability in the 20 Hz AM condition above that of the non-modulated condition. However, in re-examining the results, we recognized that the 20 Hz AM condition included an outlier that was pulling the group mean higher. We should have caught this outlier in the initial submission given that the resultant percent change for this individual is 3 standard deviations above the mean. Given the skew in the distribution, we also performed a log transform on the MEPs (which improves the normality and homoscedasticity of MEP distributions) and repeated the analysis. However, even here the participant’s results remained well outside the distribution. As such, we removed this participant and repeated all analyses. In this new analysis, there was no longer a significant difference between the 20 Hz AM and non-modulated conditions in Experiment 2. Indeed, all three true stimulation conditions (non-modulated, AM 20 Hz, AM 140 Hz) produced a similar boost in cortical excitability compared to sham. Thus, the results of Experiment 2 are consistent with those of Experiment 1, showing, in three new conditions, the efficacy of kHz stimulation on cortical excitability. But the results fail to provide evidence of an additional boost from amplitude modulation.

We posted a second bioRxiv preprint in May, 2023 (Version 2) with the corrected results for Experiment 2, along with changes throughout the manuscript given the new analyses.

Given the null results for the AM conditions, we decided to run a third experiment prior to submitting the work for publication. Here we used an alternative form of amplitude modulation (see Kasten et. al., NeuroImage 2018). In brief, we again observed a boost in cortical excitability in from non-modulated kTMP at 3.5 kHz, but no additional effect of amplitude modulation. This work is included in the third bioRrxiv preprint (Version 3), the paper that was submitted and reviewed at eLife.

(2) Statistical analysis.

Reviewer 1 raised a concern with the statistical analyses performed on aggregate data across experiments. We recognize that this is atypical and was certainly not part of an *a priori* plan. Here we describe our goal with the analyses and the thought process that led us to combine the data across the experiments.

Our overarching aim is to examine the effect of corticospinal excitability of different kTMP waveforms (carrier frequency and amplitude modulated frequency) matched at the same estimated cortical E-field (2 V/m). Our core comparison was of the active conditions relative to a sham condition (E-field = 0.01 V/m). We included the non-modulated 3.5 kHz condition in Experiments 2 and 3 to provide a baseline from which we could assess whether amplitude modulation produced a measurable difference from that observed with non-modulated stimulation. Thus, this non-modulated condition as well as the sham condition was repeated in all three experiments. This provided an opportunity to examine the effect of kTMP with a relatively large sample, as well as assess how well the effects replicate, and resulted in the strategy we have taken in reporting the results.

As a first step, we present the data from the 3.5 kHz non-modulated and sham conditions (including the individual participant data) for all three experiments in 4. We used a linear mixed effect model to examine if there was an effect of Experiment (Exps 1, 2, 3) and observed no significant difference within each condition. Given this, we opted to pool the data for the sham and 3.5 kHz non-modulated conditions across the three experiments. Once data were pooled, we examined the effect of the carrier frequency and amplitude modulated frequency of the kTMP waveform.

(3) Carry-over effects

As suggested by Reviewer 1, we will examine in the revision if there is a carry-over effect across sessions (for the most part, 2-day intervals between sessions). For this, we will compare MEP amplitude in baseline blocks (pre-kTMP) across the four experimental sessions.

Reviewer 1 also commented that mixing the single- and paired-pulse protocols might have impacted the results. While our *a priori* focus was on the single-pulse results, we wanted to include multiple probes given the novelty of our stimulation method. Mixing single- and different paired-pulse protocols has been relatively common in the non-invasive brain stimulation literature (e.g., Nitsche 2005, Huang et al, 2005, López-Alonso 2014, Batsikadze et al 2013) and we are unaware of any reports suggested that mixed designs (single and paired) distort the picture compared to pure designs (single only).

(4) Sensation and Blinding

Reviewer 2 bought up concerns about the sham condition and blinding of kTMP stimulation. We do think that kTMP is nearly ideal for blinding. The amplifier does emit an audible tone (at least for individuals with normal hearing) when set to an intensity to produce a 2 V/m E-field. For this reason, the participants and the experimenter wore ear plugs. Moreover, we played a 3.5 kHz tone in all conditions, including the sham condition, which effectively masked the amplifier sound. We measured the participant’s subjective rating of annoyance, pain, and muscle twitches after each kTMP session (active and sham). Using a linear mixed effect model, we found no difference between active and sham for each of these ratings suggesting that sensation was similar for active and sham (Fig 8). This matches our experience that kHz stimulation in the range used here has no perceptible sensation induced by the coil. To blind the experimenters (and participants) we used a coding system in which the experimenter typed in a number that had been randomly paired to a stimulation condition that varied across participants in a manner unknown to the experimenter.

Reviewer 1 asked why we did not explicitly ask participants if they thought they were in an active or sham condition. This would certainly be a useful question. However, we did not want to alert them of the presence of a sham condition, preferring to simply describe the study as one testing a new method of non-invasive brain stimulation. Thus, we opted to focus on their subjective ratings of annoyance, pain, and finger twitches after kTMP stimulation for each experimental session.

**Response to Recommendations for the Authors**

**Reviewer #1:**

Reviewer # 1 in the public review noted the possibility of carry-over effects and suggested that we compare the amplitude of the MEPS in the pre blocks across the four sessions.

Although we did not anticipate carry-over effects lasting 2 or more days, we have now conducted an analysis in which we use a linear mixed effect model with a fixed factor of Session and a random factor of Participant. The results show that there is not an effect of session [χ2(3) = 4.51, *p* = 0.211].

**Author response table 1. sa3table1:** 

Session #	MEP (Mean)	MEP (SE)
Session 1	1.12	0.12
Session 2	1.43	0.18
Session 3	1.27	0.17
Session 4	1.36	0.16

Detailed comments and some suggestions to maybe improve the writing and figures:Abstract:BioRxiv Version 1: "We replicated this effect in Experiment 2 and found that amplitude-modulation at 20 Hz produced an additional boost in cortical excitability. "BioRxiv Version 2, 3 and current manuscript: "Although amplitude-modulated kTMP increased MEP amplitude compared to sham, no enhancement was found compared to non-modulated kTMP."I am a little concerned about this history because the conclusions seem to have changed. It looks like the new data has a larger number of subjects, which could explain the divergence. Although it is generally not good practice to analyze the data at interim time points, without accounting for alpha spending. It appears that data analysis methods may have also changed, as some of the extreme points in version 1 seem to be no longer in the new manuscript (Figure 4 Sham Experiment 1).

In the public review above we explain in detail the different versions of the bioRxiv preprint and how the results changed from the first version to the current manuscript.

Introduction:"Second, the E-fields for the two methods exist in orthogonal subspaces" Can you explain what this means?

Thank you for this suggestion, we have updated the paper (pg. 4, line 78-81) by adding two sentences to explain what we mean by orthogonal subspaces and describe the consequences of this with respect to the E-fields resulting from tES and TMS. Specifically, we now comment that even if the E-fields of tES and TMS are similar in focality, they may target different populations of neurons.

"In addition, the kTMP waveform can be amplitude modulated to potentially mimic E-fields at frequencies matching endogenous neural rhythms [15]." That may be so, but reference [15] makes the exact opposite point, namely, that kHz stimulation has little effect on neuronal firing until you get to very strong fields. The paper that makes that claim is by Nir Grossman, but in my view, it is flawed as responses are most likely due to peripheral nerve (axon) stimulation there given the excessive currents used in that study. The reference to Wang and Peterchev [17] is in agreement with that by showing that you need 2 orders of magnitude stronger fields to activate neurons.

The reviewers are correct that that Ref 15 (Esmaeilpour et al, 2021), as well as Wang et al, 2023 use much higher E-fields than we target in our present study. However, our point here is that, while we cannot use our approach to apply E-fields at endogenous frequencies, we can do amplitude modulation of the kHz carrier frequency at these lower frequencies. We cited Esmaeilpour et al., (2021) because they show that high frequency stimulation with amplitude-modulated waveforms resulted in dynamic modulation at the “beating” frequency. Given we are well in subthreshold space in this paper, and well below the E-field levels in Esmaeilpour et al (2021), the open question is whether amplitude modulation at this level will be able to perturb neural activity (e.g., increase power of endogenous oscillations at the targeted frequency).

To address this concern, we modified the sentence (pg.6, lines 120-121) to now read "In addition, the kTMP waveform can be amplitude modulated at frequencies matching endogenous neural rhythms." In this way, we are describing a general property of kTMP (as well as other methods that can use high frequency signals).

I am not aware of any in-vitro study showing the effects of kHz stimulation at 2V/m. The review paper by Neudorfer et al is very good. But if I got it correctly in a quick read it is not clear that there is experimental evidence for subthreshold effects. They do talk about facilitation, but the two experimental papers cited there on the auditory nerve don't quantify field magnitudes. I would really love it if you could point me to a relevant empirical study showing the effects of kHz stimulation at 2 V/m.Perhaps all this is a moot point as you are interested in lasting (plastic) effects on MEP. For this, you cite one study with 11 subjects showing the effects of kHz tACS on MEPs [20]. I guess that is a start. The reference [21] is only a safety study, so it is probably not a good reference for that. Reference [22] also seems out of place as it is a modeling study. The effects on depression of low-intensity magnetic stimulation in references [23-26] are intriguing.

We agree with the reviewer that Ref 20 (now Ref 18: Chaieb, Antal & Paulus; 2011) is the most relevant one to cite here since it provides empirical evidence for changes in neural excitability from kHz stimulation, and in fact, serves as the model for the current study. We have retained Refs 23-26 (now Ref 19-22: Rohan et al., 2014; Carlezon et al., 2005; Rohan et al., 2004 & Dublin et al., 2019) since they also do show kHz effects on mood and removed Refs 21 (Chaieb et al., 2014) and 22 (Wang et al., 2018) for the reasons cited by the Reviewer.

Figure 1: "The gray dashed function depicts the dependence of scalp stimulation threshold upon frequency [14]." It's hard to tell from that reference what the exact shape is, but the frequency dependence is likely steeper than what is shown here, i.e. 2 mA at 10 Hz can be really quite unpleasant.

We have removed the gray dashed line given that this might be taken to suggest a discrete transition. We now just have a graded transition to reflect that the tolerance of tES is subjective. We start the shading at 2 mA for the lowest frequencies given that there is general agreement that 2 mA is well-tolerated and decrease the shading intensity as frequency increases. The general aim of the figure is not to make strong claims about the threshold of scalp discomfort for tES, but to show that kTMP can target much higher cortical E-fields within the tolerable range.

Methods:Procedures:It does not seem like double-blinding has been directly assessed.

We did not assess double blinding by directly assessing whether the participant was in a sham or active condition. We did not want to alert the participants of the presence of a sham condition after the first session of the 4-session study, preferring to simply describe the study as a test of a new method of non-invasive brain stimulation. For this reason, we opted to focus on their subjective ratings of annoyance, pain, and finger twitches after kTMP stimulation for each experimental session. These ratings did not differ between active and sham kTMP, which suggests kTMP has good potential for double blinding.

MEP data analysis: Taking the mean of log power is unusual, but I suppose the reference provided gives a good justification. Does this explain the deviation from the biorxiv v1 results?

We opted to perform a logarithmic transformation of MEP amplitudes to improve the normality and homoscedasticity of the MEP distribution. We cite three papers (Refs 50-52: Peterchev et al., 2013, Nielsen 1996a, & Nielsen 1996b) that have applied a similar approach in handling MEP data. We had not done the transformation in the first bioRxiv but opted to do so in the eLife submission based on further review of the literature. We note that the two analyses produce similar statistical outcomes once we removed the outlier discussed in the Public Review.

"Interactions were tested by comparing a model in which the fixed effects were restricted to be additive against a second model that could have multiplicative and additive effects." Not sure what this means. Why not run a full model with interactions included and read off the stats from that single model for the various factors? Should one not avoid running multiple models as one would have to correct p-values for multiple comparisons for every new test?

We used the lme4 package in R to fit our linear mixed effect models (Ref 54: Bates, Mächler, Bolker & Walker, 2015). In this package they intentionally leave out *p*-values for individual models or factors because they note there is a lack of convergence in the field about how to calculate parameter estimates in complex situations for linear mixed effect models (e.g., unbalanced designs). They suggest model comparison using the likelihood-ratio test to obtain and report *p*-values, which is what we report in the current manuscript.

We revised the text in the section *Linear Mixed Effects Models* to state that likelihood ratio tests were used to obtain *p*-values to remove any confusion.

Procedures:kTPM: Nice that fields were measured. Would be nice to see the data that established the empirical constant k.

We have expanded our discussion of how we established *k* in the Methods section. We first derived *k* using the equation E0 = kfcI based on previously published reports of the current (I) and frequency (fc) of the MagVenture Cool-B65 coil (now Refs 29-30: Deng, Lisanby & Peterchev, 2013; Drakaki, Mathiesen, Siebner, Madsen & Thielscher, 2022). We then verified this value using the triangular E-field probe to within 5% error.

Figure 3, spectrum. The placement of the fm label on the left panel is confusing. It suggests that fm was at the edge of the spectrum shown, which would not be the best way to show that there is nothing there - obviously, there isn't, but the figure could be more didactic.

Thanks for pointing this out. We modified the figure, moving the ‘fm’ label to the center of the first panel. This change makes it clear that there is no peak at the amplitude modulated frequency.

"a trio of TMS assays of cortical excitability" Can you clarify what this means?

Sorry for the confusion. The trio of TMS assays refers to the single pulse and two paired-pulse protocols (SICI - ICF). We edited the Procedure section to clarify this (pg 9, line 195-197).

Figure 2A: it would be nice to indicate which TMS blocks were single pulse and which were the two paired-pulse protocols. It is hard to keep track of it all for the three different experiments.

We have now clarified in the text (see above) that all three probes were used in each block for Experiments 1 and 2, and only the single-pulse probe in Experiment 3. We have modified the legend for Figure 2 to also provide this information.

Results:"Based on these results, we combined the data across the three experiments for these two conditions in subsequent analyses." This strikes me as inappropriate. Should not a single model have been used with a fixed effect of experiment and fixed effect of stimulation condition?

We recognize that pooling data across experiments may be atypical. Indeed, our initial plan was to simply analyze each experiment on its own (completely within-subject analysis). However, after completing the three experiments, we realized that since the sham and non-modulated 3.5 kHz conditions were included in each experiment, we had an opportunity to examine the effect of kTMP in a relatively large N study (for NIBS research). Before pooling the data, we wanted to make sure that the factor of experiment did not impact the results and our analysis showed there was no effect of experiment. Note that we did not include the factor of stimulation condition in this model because we did not want to do multiple comparisons of the same contrast (3.5 kHz compared to sham). By pooling the data before analysis of the stimulation conditions we could then focus on our two key independent variables: (1) kTMP carrier frequency and (2) kTMP amplitude modulated frequency, doing fewer significance tests to minimize multiple comparisons. The linear mixed effect (LME) model allows us to include a random effect of participant. In this way, we account for the fact that some comparisons are within subjects and some comparisons are between subjects.

The reviewer is correct that after pooling the data, we could have continued to include the factor of experiment in the LME models. This factor could still account for variance even though it was not significant in the initial test. Given this, we have now reanalyzed the data including the fixed factor of experiment in all the comparisons that contain data from multiple experiments. This has led us to modify the text in the Methods section under *Linear Mixed Effects Models* and in the Results section under *Repeated kTMP Conditions (3.5 kHz and Sham) across Experiments*. In addition, the results of the LME models have been updated throughout the Results section. We note that the pattern of results was unchanged with this modification of our analyses.

"Pairwise comparisons of each active condition to sham showed that an increase was observed following both 2 kHz ..." I suppose this is all for Experiment 1? It is a little confusing to go back and forth between combining experiments and then separate analyses per experiment without some guiding text, aside from being a bit messy from the statistical point of view.

We did not go back to performing separate analyses of the experiments after pooling the data. Once we ran the test to justify pooling the data, subsequent tests were done with the pooled data to evaluate the effects of carrier frequency and amplitude modulation.

Figure 5 is confusing because the horizontal lines with ** on top seem to refer to the same set of sham subjects, but the subjects of Experiments 2 and 3 are different from Experiment 1, so in these pairwise comparisons there is a mix of between-subject and within subject-comparison going on here. Did I get that right?

Yes – that is correct. As noted above we pooled the data after showing that there was no effect of experiment. Thus, the data for the sham and 3.5 kHz non-modulated conditions are from three different experiments. There was some overlap of subjects in Experiments 1 and Experiment 2 (Experiment 3 was all new participants). We used a linear mixed effect model so that we could account for this mixed design. Participant was always included as a random factor, which allows us to account for the fact that some comparisons are within, and some are between. Based on a previous comment, we now include Experiment as a fixed factor (see above) which provides a way to evaluate variance across the different experiments.

"We next compared sham vs. active non-modulated kTMP and found that active kTMP produced a significant increase in corticospinal excitability χ2(1) = 23.46 p < 0.001" Is this for the 3.5Hz condition?

No, that is for an omnibus comparison of non-modulated kTMP (including 2 kHz, 3.5 kHz and 5 kHz conditions) vs. sham. We have edited the paper to include the three conditions that are included as the active non-modulated kTMP conditions for clarity (pg. 22, line 463). Having observed a significant omnibus result, we continued with paired comparisons: “Pairwise comparisons of each active condition to sham showed that an increase was observed following both 2 kHz [χ2(1) = 6.90, *p* = 0.009; *d* = 0.49] and 3.5 kHz kTMP [χ2(1) = 37.75, *p* < 0.001; *d* = 0.70; Fig 5: Non-Modulated conditions]. The 5 kHz condition failed to reach significance [χ2(1) = 1.43, *p* = 0.232; *d* = 0.21].”

Paired-Pulse Assays: There are a number of results here without pointing to a figure, and at one point there is a reference to Figure 6, which may be in error. It would help to point the reader to some visual corresponding the the stats.

Thank you. This was an error on line 542. It should have read Figure 7. We have added two other pointers to Figure 7 where we discuss the absence of an effect of kTMP on SICI.

**Reviewer #2 (Recommendations For The Authors):**
I would recommend a couple of changes to the background."Orthogonal subspaces" line 78. This is a fairly formal term that has little relevance here, although the difference between scalar and vector potential-based fields is interesting to think about. If it stays, it should be mathematically supported, but it's easily rewritten to deliver the gist of it.

We have updated the paper by adding text that we hope will clarify what we mean by orthogonal subspaces (pg. 4, line 78-81). We note that we developed the math behind this statement in a previous paper (Ref # 10: Sheltraw et al., 2021). We have changed the location of the citation so that it directly follows these sentences and will provide a pointer to readers interested in the physics and math concerning orthogonal subspaces.

The statement that the scalp e-field for TES is greater than the e-field for TMS for similar cortical fields needs a little more clarification, since historically they have operated orders of magnitude apart, and it is easy to misread and trip over this statement (although it is factually true). Presenting a couple of numbers at cortical and scalp positions would help illustrate the point. That you are not considering applying TES at traditional TMS levels but rather TMS at TES values is what is initially easy to miss.

We appreciate the feedback and have updated this section to provide the reader with a better intuition of this point. We now specify that the scalp to cortical E-field ratio is approximately 18 times larger for tES compared to TMS and cite our previous paper which has much more detail about how this was calculated.

A note that the figures show scalp sensation around 1.0 V/m while the text states 0.5; cortical depths are an important thing for the reader to keep in mind.

This comment, when considered in tandem with one of the comments of Reviewer 1 led us to revise Figure 1. We removed the dashed gray line which might be taken to suggest a strict cutoff in terms of tolerability (which we did not intend). We now use shading that fades away to make the point of continuity. We have extended this down to a cortical E-field of 0.5 V/m to correspond with the text.

This is a nicely done and carefully reported experiment and I look forward to seeing more.

Thank you for your kind note!